# COMPASS: Robust Feature Conformal Prediction for Medical Segmentation Metrics

**Matt Y. Cheung, Ashok Veeraraghavan & Guha Balakrishnan**
Department of Electrical & Computer Engineering
Rice University
Houston, TX 77005, USA
`{yc130,av21,gb35}@rice.edu`

## Abstract

In clinical applications, the utility of segmentation models is often based on the accuracy of derived downstream metrics such as organ size, rather than by the pixel-level accuracy of the segmentation masks themselves. Thus, uncertainty quantification for such metrics is crucial for decision-making. Conformal prediction (CP) is a popular framework to derive such principled uncertainty guarantees, but applying CP naively to the final scalar metric is inefficient because it treats the complex, non-linear segmentation-to-metric pipeline as a black box. We introduce COMPASS, a practical framework that generates efficient, metric-based CP intervals for image segmentation models by leveraging the inductive biases of their underlying deep neural networks. COMPASS performs calibration directly in the model's representation space by perturbing intermediate features along low-dimensional subspaces maximally sensitive to the target metric. We prove that COMPASS achieves valid marginal coverage under the assumption of exchangeability. Empirically, we demonstrate that COMPASS produces significantly tighter intervals than traditional CP baselines on four medical image segmentation tasks for area estimation of skin lesions and anatomical structures. Furthermore, we show that leveraging learned internal features to estimate importance weights allows COMPASS to also recover target coverage under covariate shifts. COMPASS paves the way for practical, metric-based uncertainty quantification for medical image segmentation.

## 1 Introduction

Uncertainty quantification is of critical need in medical image analysis, a field used for decision support in high-stakes clinical diagnosis and treatment planning applications (Begoli et al., 2019; Abdar et al., 2021). A fundamental task in medical image analysis is image segmentation, the task of separating anatomical structures and lesions from each other within an image. Deep learning models, particularly U-Net variants (Ronneberger et al., 2015; Isensee et al., 2021), have achieved state-of-the-art performance in medical image segmentation. In practice, the outputs of these models ("segmentation maps") are often treated as an intermediate result that are then used to automatically derive downstream metrics of interest (known as "radiomics"), such as the areas/volumes or texture patterns of specific anatomic regions (Figure 1, left). These derived metrics are then used for decision support to guide clinicians in diagnosis and treatment.

Conformal prediction (CP) has emerged as a popular, statistically principled uncertainty quantification framework of choice in machine learning, providing guarantees without restrictive distributional assumptions (Vovk et al., 2005; Shafer & Vovk, 2008; Fontana et al., 2023; Papadopoulos et al., 2002; Angelopoulos & Bates, 2021). While well-studied in the context of typical prediction tasks involving scalar output variables, CP is less explored for tasks such as medical image segmentation, in which the output variables are images. Existing CP methods for segmentation typically focus on deriving bounds for *pixel-level errors* (Mossina et al., 2024; Mossina & Friedrich, 2025; Brunekreef et al., 2024; Angelopoulos et al., 2022), which, while useful for understanding variations of local segmentation contours, may yield meaningless or misaligned intervals for downstream derived metrics. On the other hand, a recent study shows that treating the segmentation-to-metric

pipeline as a black box and performing CP directly on the output metric space yields intervals that are well-aligned to the metrics (by construction), but are also often inefficient (i.e., large) because the internal biases of the pipeline are not exploited in the vanilla CP formulation (Cheung et al., 2025).

To achieve more efficient intervals, a promising direction is to leverage the powerful inductive biases of neural networks by performing CP on their intermediate representations. Feature Conformal Prediction (FCP) (Teng et al., 2022; Tang et al., 2024; Chen et al., 2024) shows that by working in a semantic feature space, it is possible to generate provably tighter prediction intervals. However, the FCP algorithm requires solving a complex optimization to find the closest adversarial feature vector for each data point, which is computationally prohibitive for high-dimensional feature spaces of typical CNN and transformer architectures which are the workhorses of modern medical image segmentation.

To bridge this gap, we introduce **Conformal Metric Perturbation Along Sensitive Subspaces (COMPASS)**, a framework to perform feature CP in a tractable manner to generate valid and efficient prediction intervals for any (differentiable) metric derived from the output of a neural network. The core concept behind COMPASS is to linearly perturb the features outputted by the network at a particular layer along data-specific directions that are highly sensitive to the metric of interest (Figure 1, center). To make this process tractable for typical neural network layers with a large number of features, we propose computing a low-dimensional manifold of any given layer by applying principal component analysis (PCA) on the gradients of the output metric with respect to each of the layer's features. This manifold represents the principal directions of sensitivity of the output with respect to that layer. We prove that linear perturbations in the latent space achieve marginal coverage in the output metric space under exchangeability, i.e., for a fresh test point, the ground truth will be contained within the prediction intervals generated by linear feature perturbations. Furthermore, we show that a simple weighted variant of COMPASS may be used to correct for covariate shifts.

We evaluated COMPASS on public datasets for four medical image segmentation tasks: colorectal cancer in histopathology images (EBHI) (Hu et al., 2023), skin lesion segmentation (HAM10000) (Tschandl et al., 2018), thyroid nodule segmentation in ultrasound images (TN3K) (Gong et al., 2021), and polyp segmentation in endoscopic images (Kvasir) (Jha et al., 2019). Results show that while standard CP methods achieve valid coverage, they often produce unnecessarily wide prediction intervals. On the other hand, COMPASS finds semantically meaningful directions in the latent space that correspond to monotonic metric changes, resulting in efficient intervals. Furthermore, the weighted extension of COMPASS recovers target coverage under covariate shifts and is the most efficient across all weighted baseline methods. COMPASS paves the way for practical, metric-based uncertainty quantification for medical image segmentation.

## 2 METHOD

### 2.1 THEORETICAL COVERAGE UNDER LINEAR LATENT PERTURBATIONS

To provide rigorous uncertainty quantification for the metric $Y$, we consider *linear perturbations in latent space* along a direction $\Delta \in \mathcal{Z}$ and define intervals in the metric space. We first formalize the nestedness condition, which is a fundamental requirement for the validity of any conformal procedure (Vovk et al., 2005; Shafer & Vovk, 2008).

**Definition 1** (Nestedness). *Let $S_\beta(x)$ be a family of prediction sets for an input $x \in \mathcal{X}$, parameterized by $\beta \geq 0$. The family $\{S_\beta(x)\}_{\beta \geq 0}$ is said to be* nested *for $x$ if $\beta_1 \leq \beta_2 \implies S_{\beta_1}(x) \subseteq S_{\beta_2}(x)$.*

This condition guarantees that a larger perturbation magnitude yields a larger (or equal-sized) prediction set, which is necessary for the quantile-based coverage proof. While standard CP methods satisfy this trivially, it becomes a non-trivial condition for deep feature spaces. We now present our main theorem. Crucially, we construct our prediction sets $S_\beta(x)$ in a way that *guarantees nestedness by definition*, thereby ensuring the validity of the conformal procedure.

**Theorem 1** (Split-Conformal Coverage under Linear Latent Perturbations). *Let $(X_i, Y_i)_{i \geq 1}$ be exchangeable random pairs with $X_i \in \mathcal{X}$ and $Y_i \in \mathbb{R}$, and split the data into a training set $\bar{D}_{\mathrm{tr}}$ and a calibration set $D_{\mathrm{cal}} = \{(X_i, Y_i)\}_{i=1}^n$. Using $D_{\mathrm{tr}}$, fit a segmentation model with decoder $g : \mathcal{Z} \to \mathcal{S}$, and let $\hat{z}(x) \in \mathcal{Z}$ denote the latent vector computed deterministically by the trained encoder for in-*

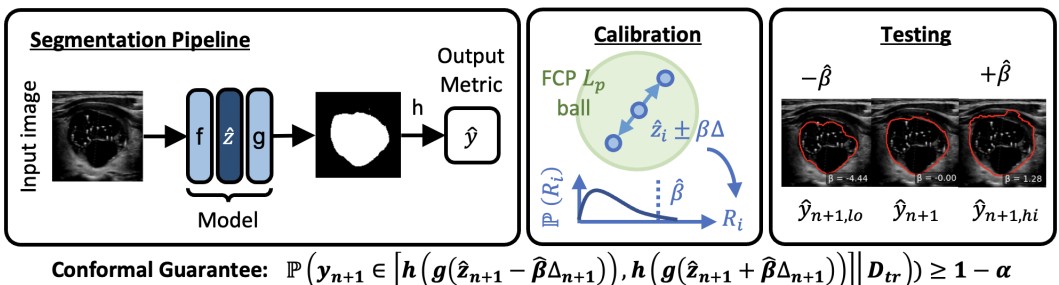

Conformal Guarantee: $\mathbb{P}\left(y_{n+1} \in \left[h\left(g(\hat{z}_{n+1} - \widehat{\beta}\Delta_{n+1})\right), h\left(g(\hat{z}_{n+1} + \widehat{\beta}\Delta_{n+1})\right)\right] \middle| D_{tr}\right) \geq 1 - \alpha$

Figure 1: **Overview of COMPASS.** (Left) A medical image segmentation network predicts a segmentation map" from an input image. We conceptually decompose this network into a function $f$ which maps the image to latent features $\hat{z}$, and a function $g$ that maps $\hat{z}$ to the output map. The map may then be used to compute a (differentiable) downstream metric $\hat{y}$ via the function $h$. (Center) We linearly perturb calibration features $\hat{z}_i$ in a sample-specific direction $\Delta_i$ to find the scores $R_i$. The scores are used to find the conformal quantile $\hat{\beta}$. (Right) At test time for subject $n+1$, we perturb the features $\hat{z}_{n+1}$ in the direction $\Delta_{n+1}$ with magnitude $\hat{\beta}$. By Theorem 1, our interval construction is guaranteed to be nested. Therefore, under the assumption of exchangeability, the resulting prediction interval achieves marginal coverage (bottom).

*put $x$. Let $h : \mathcal{S} \to \mathbb{R}$ be a measurable metric, and let $\Delta \in \mathcal{Z}$ be any measurable direction that depends only on $D_{\text{tr}}$.*

*For $x \in \mathcal{X}$ and $\beta \geq 0$, define the metric function along this direction as:*

$$m_x(b) := (h \circ g)(\hat{z}(x) + b\Delta), \quad \text{for } b \in \mathbb{R}. \tag{1}$$

*We define the prediction set $S_\beta(x)$ as the range of the metric function over the perturbation interval $[-\beta, +\beta]$:*

$$S_\beta(x) := \left[\min_{b \in [-\beta, +\beta]} m_x(b), \max_{b \in [-\beta, +\beta]} m_x(b)\right]. \tag{2}$$

*By construction, this guarantees that the family $\{S_\beta(x)\}_{\beta \geq 0}$ is nested (Definition 1). For each calibration pair $(X_i, Y_i)$, define the non-conformity score:*

$$R_i := \inf\{\beta \geq 0 : Y_i \in S_\beta(X_i)\} \in [0, \infty], \tag{3}$$

*and let $\hat{\beta}$ be the $\lceil(1 - \alpha)(n + 1)\rceil$-th smallest value among $\{R_1, \ldots, R_n\}$, where $\alpha \in (0, 1)$ is the user-specified mis-coverage level.*

*Then, for a fresh test pair $(X_{n+1}, Y_{n+1})$, the prediction set $S_{\hat{\beta}}(X_{n+1})$ satisfies*

$$\mathbb{P}\big(Y_{n+1} \in S_{\hat{\beta}}(X_{n+1}) \mid D_{\text{tr}}\big) \geq 1 - \alpha. \tag{4}$$

*Proof sketch. Exchangeability of $(X_i, Y_i)$ ensures that the rank of the test score $R_{n+1}$ among $\{R_1, \ldots, R_n, R_{n+1}\}$ is uniformly distributed. By the construction of $S_\beta(x)$ as the range $[\min(\cdot), \max(\cdot)]$, the nestedness condition is satisfied by definition. Therefore, by construction of $\hat{\beta}$, the standard conformal guarantee holds. See Appendix A for full proof.*

**Intuition.** Our prediction set $S_\beta(x)$ is defined as the metric *range* over the perturbation interval, which satisfies the nestedness condition required for CP. Our construction forms a *conservative envelope* that explicitly accounts for any non-monotonic metric behavior within the perturbation interval. Our non-conformity score $R_i$ is then the minimal perturbation magnitude $\beta$ required for this envelope to contain the ground truth metric $Y_i$.

## 2.2 COMPASS: CONFORMAL METRIC PERTURBATION ALONG SENSITIVE SUBSPACES

The proposed framework COMPASS is built on the insight that we can directly calibrate a downstream metric by perturbing a model's representations along specific, data-dependent directions, $\Delta$.

As our theoretical guarantee in Theorem 1 applies to any measurable perturbation direction, the key to an efficient method lies in choosing a direction that is highly sensitive to the metric of interest. We explore two natural choices for this representation space for segmentation: the model's final output logits and a deeper internal feature layer.

The first and simplest approach (**COMPASS-L**) operates on the model's final logits. A uniform, scalar shift applied to this tensor directly modulates the model's overall confidence before the final activation function. This is equivalent to defining the sensitive direction, $\Delta_i$, as a tensor of ones. A richer alternative is to perturb an internal feature representation $\hat{z}$, as done in FCP (Teng et al., 2022). However, a naive search for an optimal perturbation in high-dimensional spaces is computationally intractable. Furthermore, arbitrary perturbation directions may cause the metric to change in an erratic or non-smooth ways, resulting in unnecessarily wide and inefficient intervals.

To overcome this, we propose a data-driven method (**COMPASS-J**) to identify a low-dimensional sensitive subspace that is globally effective[1]. Given a metric function in Equation 1 that is differentiable, we first compute Jacobians of the metric $\hat{y}$ with respect to $\hat{z}$ to provide a local, linear map of the metric's sensitivity with respect to the features for each training sample:

$$J_i := \frac{d\,h(g(\hat{z}_i))}{d\hat{z}_i} \in \mathbb{R}^{C \times D_1 \times \cdots \times D_3},$$

where $\hat{z}_i = f(x_i)$, $C$ is the number of channels, and $D_i$ is the $i$-th spatial dimension. Because the full spatial Jacobian is often too high-dimensional to be practical, we *sum* the spatial dimensions and apply Principal Component Analysis (PCA) to the set of these vectors from the training set, $\{\mathcal{J}_i\}_{i \in D_{\mathrm{tr}}}$, and select the matrix of the top $L$ eigenvectors $V_L \in \mathbb{R}^{C \times L}$.

For any given sample, we find its sensitive direction $\mathbf{d}_i$ by projecting its sensitivity vector onto this learned subspace, and normalization it to produce direction vector $\Delta_i$:

$$\mathbf{d}_i = V_L V_L^T \mathcal{J}_i, \quad \Delta_i = \mathbf{d}_i / \|\mathbf{d}_i\|_2.$$

## 2.3 COMPASS CALIBRATION AND INFERENCE

The goal of COMPASS is to find the smallest symmetric perturbation magnitude $\beta$ such that the entire interval $S_\beta(x)$ contains the ground truth value $y$. In general, computing $S_\beta(x)$ requires finding the minimum and maximum of the perturbed metric response $m_{x_i}(b), b \in [-\beta, \beta]$. When the metric response is *non-monotonic*, computing the extrema requires a full discretized sweep across the perturbation range. Performing this sweep at every step during calibration makes the envelope computation expensive because each candidate $\beta$ requires many forward passes to map out the complete response curve for each sample.

However, when $m_{x_i}(b)$ is *monotonic* in $b$, the conservative envelope collapses to evaluations at the *endpoints*: $S_\beta(x_i) = \big[m_{x_i}(-\beta),\ m_{x_i}(+\beta)\big]$. Thus, the interval can be computed with only two forward passes, giving a practical and efficient implementation of COMPASS. A perturbation sweep is still required, but only once per sample, solely to verify monotonicity rather than to repeatedly compute the envelope during calibration. As we will see in Figures 2 and 12, this monotonicity condition holds across all of our experiments.

Once $S_\beta(x_i)$ can be evaluated, either via the conservative full sweep or the endpoint method, the non-conformity scores $\{R_i\}_{i=1}^n$ follow directly from Equation 3. The conformalized quantile is:

$$\hat{\beta} = Q\left(\{R_i\}_{i=1}^n, \frac{\lceil (1-\alpha)(n+1) \rceil}{n}\right),$$

and at test time we return the interval $S_{\hat{\beta}}(x_{n+1})$.

For a full outline of the algorithms, refer to Appendix C. Furthermore, while symmetric perturbations are effective when the metric responds similarly to positive and negative perturbations, we often observe an asymmetric relationship when the metric responds differently in positive and negative directions. In such cases, an asymmetric calibration, where we find separate non-conformity scores for the upper and lower bounds, is necessary to construct a more adaptive and efficient interval. We discuss this asymmetric version of COMPASS and provide an equivalent algorithm in Appendix B.

---

[1]Our use of principal directions to restrict the perturbation search space is conceptually related to Belhasin et al. (2023), where authors employed principal directions to construct prediction sets for inverse problems.

## 2.4 Weighted COMPASS for Distribution Shifts

While Theorem 1 provides a robust guarantee of marginal coverage, its validity rests on the critical assumption that the calibration and test data are exchangeable. In many real-world applications, such as medical image segmentation, this assumption is frequently violated due to the variations in how the data is collected, processed, and interpreted. Under such distribution shifts, the unweighted quantile $\hat{\beta}$ is no longer guaranteed to provide the target coverage level on the test set, leading to systematic undercoverage and unreliable prediction intervals.

To address this limitation, we employ Weighted Conformal Prediction (WCP) (Tibshirani et al., 2019; Barber et al., 2023) to restore the coverage guarantee by re-weighting the calibration samples. The weight for a calibration sample $X_i$ is ideally the density ratio $w(X_i) = p_{\text{test}}(X_i)/p_{\text{cal}}(X_i)$. In practice, this ratio is unknown and typically estimated by training an auxiliary classifier $\mathcal{A} : \mathcal{X} \to [0, 1]$ to distinguish between samples from the calibration and test sets. The predicted probability $\hat{p}(x) = \mathcal{A}(x)$ that a sample $x$ belongs to the test set is then used to compute the weights, effectively adjusting the calibration procedure to account for the distribution shift. We now extend Theorem 1 to the weighted setting, providing a formal coverage guarantee under distribution shift. The setup remains identical.

**Proposition 1** (Validity of Weighted COMPASS under Covariate Shift). *Let $D_{\text{cal}} = \{(X_i, Y_i)\}_{i=1}^{n}$ be $n$ exchangeable pairs drawn from a distribution $P_{\text{cal}}$, and let $(X_{n+1}, Y_{n+1})$ be a fresh test pair from a potentially different distribution $P_{\text{test}}$. Let the non-conformity scores $R_i = R(X_i, Y_i)$ be computed as described in Theorem 1. Let the weights $w_i = w(X_i)$ be the true density ratio $p_{\text{test}}(X_i)/p_{\text{cal}}(X_i)$. Let $\hat{\beta}_{\text{w}}$ be the weighted $(1-\alpha)$-quantile of the calibration scores $\{R_1, \ldots, R_n\}$ with corresponding weights $\{w_1, \ldots, w_n\}$, defined as*

$$\hat{\beta}_{\text{w}} := \inf\left\{\beta \geq 0 : \frac{\sum_{i=1}^{n} w_i \mathbf{1}\{R_i \leq \beta\}}{\sum_{j=1}^{n} w_j} \geq 1 - \alpha\right\}.$$

*Then, for the fresh test pair $(X_{n+1}, Y_{n+1}) \sim P_{\text{test}}$, the prediction set $S_{\hat{\beta}_{\text{w}}}(X_{n+1})$ satisfies*

$$\mathbb{P}\big(Y_{n+1} \in S_{\hat{\beta}_{\text{w}}}(X_{n+1}) \mid D_{\text{tr}}, D_{\text{cal}}\big) \geq 1 - \alpha.$$

*Proof Sketch.* This is a direct application of WCP (Tibshirani et al., 2019). The validity of this framework applies to any valid non-conformity score. Our scores are valid as they are from a deterministic function of the samples and the pre-trained model. When the weights represent the true density ratio, the weighted empirical distribution of the calibration scores is an unbiased estimator of the test score distribution. Thus, the coverage guarantee from the original theorem directly applies. □

In practice, the true oracle is not available, and the coverage guarantee holds *approximately*, with the quality of the approximation depending on the accuracy of the density ratio estimates. To correct for the induced distribution shift, we explore three correction strategies: 1) Class: available ground truth class labels as features and weights are computed directly from the known class prevalences, 2) Latent: Model's latent representations summed on the spatial dimension as features (Lambert et al., 2024), and 3) Jacobian: model's internal geometric sensitivity (jacobians) summed as features.

## 3 Experiments

We evaluated COMPASS across four medical image segmentation tasks: 1) segmentation on H&E histopathology images from the EBHI dataset (H&E) (Hu et al., 2023), 2) skin lesion segmentation on dermoscopic images from the HAM10000 dataset (Skin Lesion) (Tschandl et al., 2018), 3) thyroid nodule segmentation from the TN3K dataset (Nodule) (Gong et al., 2021), and 4) gastrointestinal polyp segmentation (PolyP) (Jha et al., 2019). We trained all models using the standard U-Net architecture from MONAI (Cardoso et al., 2022; Kerfoot et al., 2018). We focused on segmented object size (area) as the downstream clinical metric of interest. Object size is among the most widely adopted quantitative biomarkers across diverse clinical applications (Smith et al., 2003; O'Connor et al., 2008). For COMPASS-J, which requires a differentiable metric function, we compute area by applying a soft thresholding (a sigmoid function) to the output logits and then summing

the resulting probability map. We repeated each experiment over 100 randomized splits to compute average coverage and interval sizes. We provide details on architecture, preprocessing, training, calibration, and testing details in Appendix E. We considered three types of calibration strategies:

**1. Output-space calibration:** These include split conformal prediction (SCP) (Lei et al., 2018), Conformalized Quantile Regression (Output-CQR) (Romano et al., 2019), and locally adaptive conformal prediction (Local CP) (Lei et al., 2018; Papadopoulos et al., 2008; 2011).

**2. End-to-end calibration:** We use Conformalized Quantile Regression (E2E-CQR) (Lambert et al., 2024) on a model trained to directly produce pixel-wise lower and upper bounds using the Tversky loss (Salehi et al., 2017).

**3. Feature-space calibration:** These include COMPASS-L, which directly calibrates the model's final pre-activation outputs (logits), and COMPASS-J, which constructs prediction intervals by perturbing latent features along dominant directions. As the strong monotonicity is empirically validated in Figures 2 and 12, all our experiments utilize the practical endpoint algorithm. We omitted the original FCP method (Teng et al., 2022) from our main empirical comparison because we found its core adversarial search procedure for finding non-conformity scores to be computationally intractable for large feature spaces and to fail to reliably converge, preventing the computation of defined scores. We instead offer a comparison against a conceptual oracle benchmark for FCP in Table 3: because FCP prediction set is defined by an $L_p$ ball in the latent space (Teng et al., 2022) with bounds computed using Linear Relaxation based Perturbation Analysis (LiRPA) (Xu et al., 2020), we empirically find the minimal radius of the $L_p$ ball, which, when propagated through the decoding function using LiRPA, achieves the target coverage for the final metric.

## 3.1 STANDARD CONFORMAL PREDICTION

First, we qualitatively find that COMPASS finds semantically meaningful directions in the latent space that correspond to monotonic changes in metric (Figure 2) for all applications. Next, we compare interval size across methods and datasets for $\alpha = \{0.15, 0.1, 0.15\}$ (Table 1). We find that output space methods (SCP, CQR, Local) generally achieve valid coverage (Table 2) but yield longer intervals. End-to-end CQR provides tighter intervals. COMPASS (-J and -L) achieves valid coverage with a significantly reduced interval size compared to E2E-CQR. Moreover, we observe an efficiency benefit of using deeper representations (COMPASS-J) compared to logits (COMPASS-L) for the majority of datasets. COMPASS-J generally achieves the shortest interval lengths overall. We also find that COMPASS maintains similar calibration stability as baseline methods (Figure 7). This highlights the efficiency of COMPASS methods.

Table 1: **COMPASS achieves more efficient interval sizes compared to baseline methods across different target coverages.** For 4 datasets and 100 random splits, we compare interval lengths at $\alpha = \{0.15, 0.1, 0.05\}$. We show the output space, end-to-end, and feature calibration methods in red, blue, and green. The shortest mean interval lengths are bolded. For empirical coverages, see Table 2.

| | | Interval Size (Pixels$^2$, Mean±Standard Deviation) | | | | | |
|---|---|---|---|---|---|---|---|
| Dataset | $\alpha$ | COMPASS-J | COMPASS-L | E2E-CQR | Local | Output-CQR | SCP |
| H&E | 0.05 | 4637±630 | **4408±432** | 5121±651 | 6297±722 | 5646±358 | 5542±676 |
| H&E | 0.10 | 3160±336 | **3139±375** | 3433±293 | 4223±558 | 3879±369 | 3509±333 |
| H&E | 0.15 | **2320±252** | 2354±146 | 2679±199 | 3175±291 | 2819±207 | 2550±196 |
| Skin Lesion | 0.05 | **1657±80** | 1689±83 | 2569±195 | 3797±237 | 10857±65 | 3273±229 |
| Skin Lesion | 0.10 | **1179±53** | 1208±58 | 1351±75 | 2433±101 | 4581±36 | 1813±127 |
| Skin Lesion | 0.15 | **934±30** | 956±33 | 943±47 | 1865±50 | 2634±44 | 1124±77 |
| Nodule | 0.05 | **3257±210** | 3394±280 | 4150±265 | 3981±202 | 7481±46 | 4589±431 |
| Nodule | 0.10 | **2444±174** | 2510±180 | 2788±154 | 3311±133 | 5603±57 | 3076±200 |
| Nodule | 0.15 | **2016±143** | 2082±142 | 2150±164 | 2877±111 | 4032±64 | 2408±154 |
| PolyP | 0.05 | **5489±575** | 6376±769 | 9162±804 | 12394±2577 | 8163±722 | 8570±766 |
| PolyP | 0.10 | **4056±293** | 4397±469 | 6184±616 | 5965±1011 | 4981±675 | 6237±564 |
| PolyP | 0.15 | **3394±290** | 3686±361 | 4528±487 | 4463±481 | 3913±326 | 4504±366 |

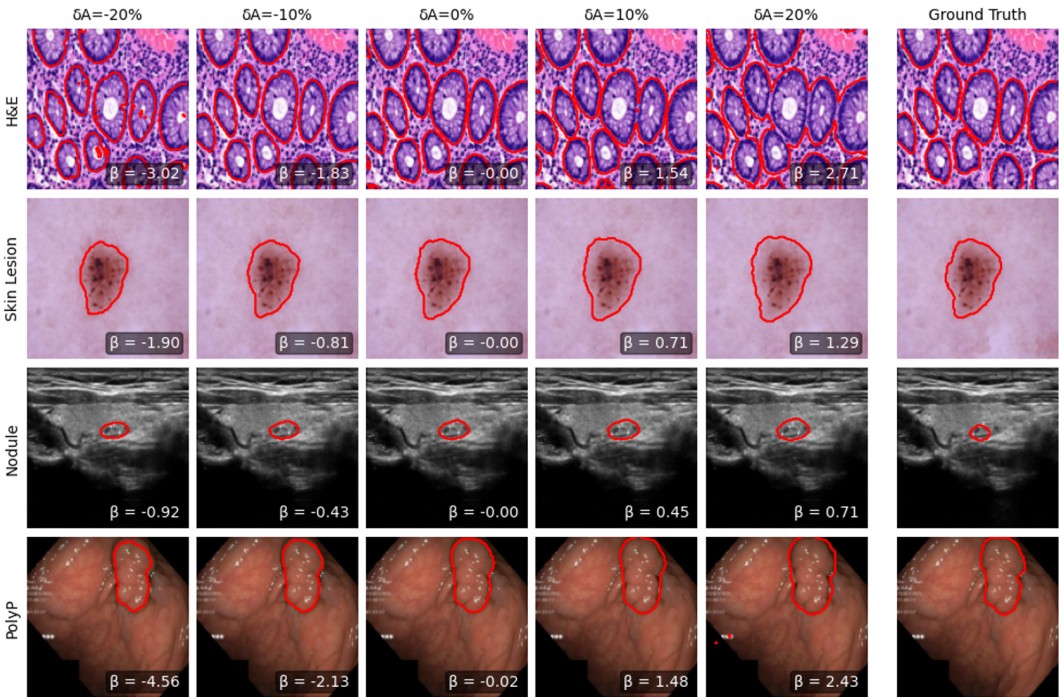

Figure 2: **Visual verification of monotonicity to justify Endpoint-COMPASS.** As latent features are shifted along the COMPASS-J direction $\Delta$, the induced segmentation volumes (red contours) monotonically expand. This is the key justification for using our efficient Endpoint-COMPASS in our experimental setup, as it demonstrates mathematical equivalence with the rigorous Envelope-COMPASS. We show a sample from each dataset with perturbation magnitudes $\beta$ targeted at -20%, -10%, 0% (original prediction), +10%, and +20% change in area ($\delta A$). We provide a plot of all metric responses on the testing datasets in Figure 12 and more visual examples in Appendix G.

## 3.2 WEIGHTED CONFORMAL PREDICTION

To empirically validate the effectiveness of WCP under distribution shift, we design an experiment with a controlled, adversarial label shift for the H&E and Skin Lesion datasets. We adopted a dataset-specific partitioning strategy based on sample availability. For H&E, we trained on a restricted subset to prevent data exhaustion, ensuring sufficient minority samples remained. Conversely, for Skin Lesion, we utilized the full dataset, inducing shift by systematically reallocating the class distributions. For H&E, we induced a shift from "easy" to "hard". We allocated 40% of the Adenocarcinoma samples to the calibration set and the remaining 60% to the test set. This results in a test distribution dominated by difficult samples, leading to baseline undercoverage. For Skin Lesion, we induced a shift from "hard" to "easy". We allocated 30% of the Melanocytic Nevi (majority/easy) samples to the calibration set, forcing the calibration set to be composed primarily of diverse, difficult lesions. The remaining 70% of the Nevi were allocated to the test set. This resulted in a calibration set that was significantly more difficult than the test set, leading to baseline overcoverage. See Appendix E for more details.

For output-space calibration methods, we used weights based on the ground truth class labels, which we consider as approximately "oracle" weights. Note that the theoretically perfect oracle weights are defined by the density ratio, which precisely corrects for the change in the full, high-dimensional distribution of features between the calibration and test sets. For end-to-end and feature space calibration, we used the latent features (Lambert et al., 2024; Woodland et al., 2023; Anthony & Kamnitsas, 2023) and Jacobians to train auxiliary classifiers using gradient boosting machines (Ke et al., 2017) to distinguish between calibration and test sets. For end-to-end, we do not use Jacobian weights consistent with prior work (Lambert et al., 2024).

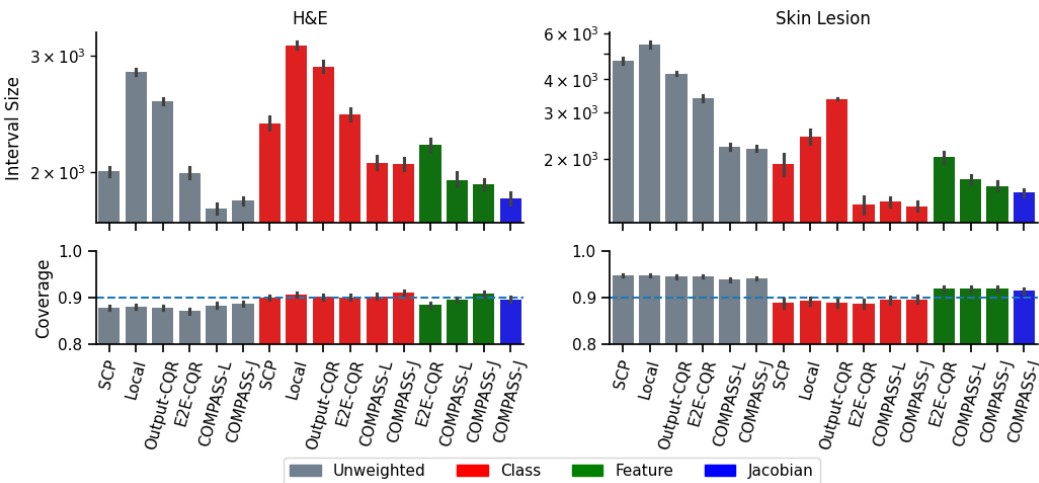

Figure 3: **COMPASS achieves the most efficient interval sizes under covariate shifts.** We show results for two datasets and compare weighting methods for 100 adversarial splits that maintain the same covariate shift for $\alpha = 0.1$. For H&E, we increased the proportion of "hard" samples in the test set. For Skin Lesion, we decreased the proportion of "hard" samples in the test set. We find that COMPASS methods achieve valid coverage and the most efficient intervals in each weighting method. We show the 95% confidence intervals.

Our results (Figure 3) demonstrate that both the choice of weighting information and the *choice of calibration layer* are critical for achieving robust coverage under these covariate shifts. Simple strategies, such as class-weighting, do not universally recover target coverage; unweighted methods (grey) fail on the H&E dataset, while class-weighted methods (red) fail on Skin Lesion. Furthermore, not all feature-based methods are robust. COMPASS-L and E2E-CQR with feature weighting still fail to maintain coverage under covariate shift on the H&E dataset. In contrast, the COMPASS-J variants with both feature-weighting and Jacobian-weighting were the only methods to consistently maintain the target coverage across both covariate shifts. Among the methods that proved empirically valid, the COMPASS-J variants were also the most efficient. This suggests that 1) the model's deep features or their Jacobians provide a richer, more adaptive signal for difficulty than simpler class labels or logits, and 2) this deep-layer signal is essential, as methods relying on shallower layers (like COMPASS-L and E2E-CQR) were not robust even when using the same feature-weighting. Our results are tabulated in Table 5.

### 3.3 EMPIRICAL ANALYSIS OF STATISTICAL EFFICIENCY

Unlike the original FCP framework, which provides a formal inequality under mild assumptions due to its reliance on the simple and convex $\mathcal{L}_p$ norm as a score, a similar theorem for COMPASS is intractable, because the COMPASS score $R_{COMPASS}$ is defined implicitly through a search process that depends on a highly non-linear and non-convex segmentation-to-metric pipeline. Instead, we investigated the reason for efficiency gains of COMPASS compared to output-space methods by finding the relationship between COMPASS scores and SCP scores $R_{SCP}$.

The statistical efficiency of COMPASS is a direct consequence of a compressive power-law relationship between feature-space scores ($R_{COMPASS}$) and output-space errors ($R_{SCP}$). A log-log plot of these quantities reveals a linear relationship with a scaling exponent slope$< 1$ (Figure 4, top). This sub-linear scaling is the direct mechanism for the tail-end compression of the score distribution (Figure 4, bottom). Thus, the largest output-space errors are systematically mapped to disproportionately smaller feature-space scores. A distribution with a compressed tail necessarily has a smaller quantile, which is the fundamental mechanism that enables COMPASS to produce tighter, more statistically efficient prediction intervals.

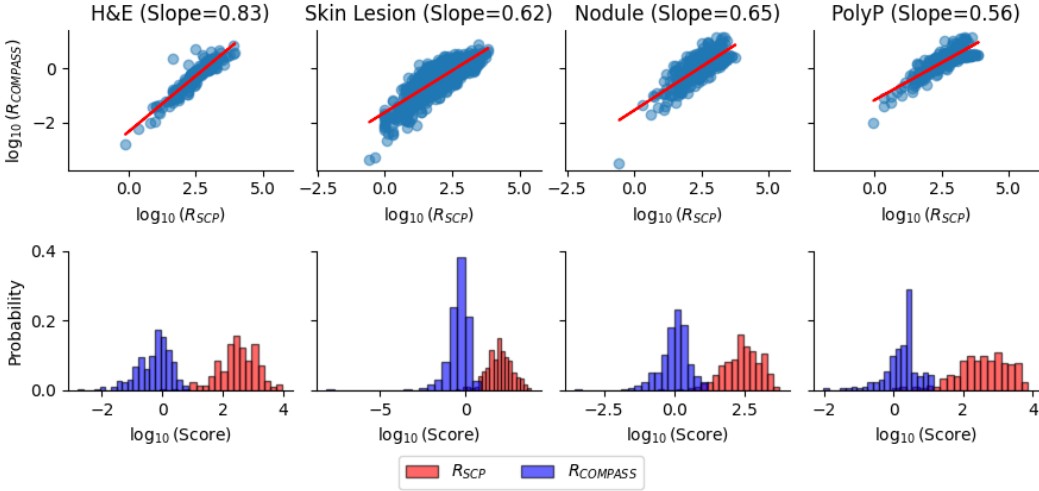

Figure 4: **The statistical efficiency of COMPASS is driven by a compressive power-law relationship between latent $R_{COMPASS}$ and output $R_{SCP}$ space scores.** As $R_{SCP}$ increases, the required latent space perturbation magnitude increases, but at a progressively slower rate since the scaling exponent (slope) is $< 1$ (Figure 4, top). This concave and sub-linear scaling is the direct cause of a tail-end compression of the score distribution (bottom). Thus, the long-tail errors are systematically transformed to much smaller feature-space scores.

## 4 DISCUSSION

We introduced COMPASS, a unified framework to generate efficient, metric-based prediction intervals by leveraging the inductive biases of neural networks. Under the assumption of exchangeability, COMPASS creates practical feature-space calibration by perturbing a model's intermediate representations along low-dimensional subspaces maximally sensitive to the target metric. Across four medical segmentation tasks for area estimation, COMPASS achieves valid coverage while producing significantly tighter intervals than traditional output-space and end-to-end baselines for both standard and weighted CP. We discuss several further points below.

COMPASS has several attractive properties. First, **it produces instance-adaptive intervals.** For COMPASS-J, the perturbation direction for any new sample is found by projecting its Jacobian onto the principal component and reconstructing the direction. Although COMPASS-L applies a uniform scalar shift, it achieves adaptivity because this shift interacts with the spatially varying logit distribution unique to each input, resulting in instance-specific adjustments to the segmentation boundary. Second, **COMPASS performs better than naive output-space calibration.** In tasks like segmentation, the final metric is a complex non-linear function of the model's output (the pixel mask). Simply adding a margin to a final object size prediction is a crude approximation that fails to capture how the object size is actually derived from the underlying segmentation. In contrast, COMPASS exploits the model's internal structure and spatial understanding to directly manipulate and calibrate the metric. COMPASS has more information and degrees of freedom, which allows it to achieve more efficient interval sizes. Third, **COMPASS performs better than methods trained using pixel-level losses** such as E2E-CQR. Such approaches optimize a *proxy* (pixel quantiles) for the downstream metric, that may or may not translate into accurate intervals for the final metric. On the other hand, COMPASS is directly calibrated to the final metric, and is therefore more applicable to practical clinical use cases.

Finally, **COMPASS is naturally applicable to various segmentation architectures.** We also present the interval lengths and coverage for SegResNet (Myronenko, 2018) in Appendix D. COMPASS produces the most efficient interval lengths on the majority of datasets across $\alpha = \{0.05, 0.1, 0.15\}$ for SegResNet.

**Limitations.** COMPASS's performance is fundamentally dependent on the quality of the pre-trained model's representations. Our practical algorithm implementation presupposes that the model has learned an inductive bias where the chosen feature space has a reasonably monotonic and sensitive control over the downstream metric. If the features are poorly aligned with the metric, the resulting perturbation direction may be inefficient or non-monotonic, leading to wide or invalid intervals. If so, we recommend using the full discretized sweep across the perturbation range to find the minimum and maximum values. Furthermore, as mentioned in prior work (Lambert et al., 2024), WCP with feature or jacobian weights, is effective for moderate distribution shifts where the calibration and test distributions have significant overlap in the feature space. However, for large shifts, the estimated weights may be inaccurate, which can compromise the validity of the final interval.

## 4.1 PRACTICAL CONSIDERATIONS

While COMPASS-L may be applicable to UNet-style architectures and object size, other applications may require different setups. Careful design of the different components of COMPASS, specifically COMPASS-J, will significantly enhance its applicability and performance. We provide a list of practical considerations to keep in mind when using COMPASS.

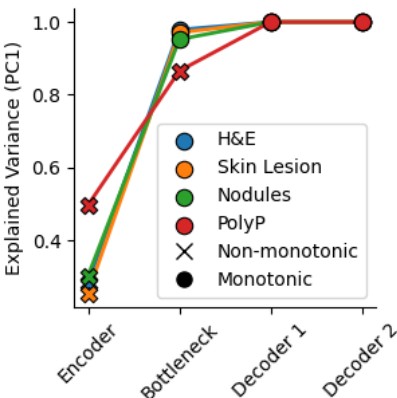

**Optimal Feature Layer.** The choice of the feature layer is critical. We find that the optimal layer can be selected empirically by measuring the *explained variance* of the first principal component ($L = 1$) of the Jacobians. High explained variance in the first principal component indicates that the local gradients of the target metric are globally aligned across the samples. Consequently, there is a high chance that a linear traversal along this principal direction results in a consistent, monotonic shift in the output metric for the entire dataset, as the projection scalar maintains a consistent sign. As shown in Figure 5, layers closer to the logits capture the most variance related to object size, validating our use of COMPASS-J. We recommend finding layers with high explained variance of the Jacobians and verify monotonicity with a perturbation sweep (Algorithm 2).

Figure 5: **Explained variance is a good indicator of monotonicity.** For our 4 datasets, we plot the first principal component's explained variance against the feature layer used for COMPASS-J. Monotonicity and non-monotonicity is indicated by ● and ✗.

**Number of PCA Components.** The number of eigenvectors $L$ controls the dimensionality of the perturbation subspace. We find that for the segmentation area metric, the first principal component already captures over 90% of the metric's variance (Figure 5). Adding more components does not significantly improve interval tightness (Figure 9). Therefore, we recommend $L = 1$ for this task, although this may be task-dependent.

**Spatial Entanglement vs Global Semantic Alignment.** We observe that the explained variance of the Jacobians is significantly higher when computed on spatially aggregated features compared to raw flattened features (Figure 10). This results from the translation-variant nature of the flattened space, where sensitivity is tied to specific pixel coordinates that vary with object position, leading to orthogonal gradients across samples. In contrast, summing the spatial dimensions isolates the channel-wise semantic sensitivity, which remains consistent across samples. We recommend summing the spatial dimensions of the Jacobians before applying dimensionality reduction techniques.

**Computationally Efficiency.** COMPASS maintains high computational efficiency, particularly when the metric response is monotonic. The jacobian computation is inexpensive as gradients are calculated with respect to the scalar metric rather than the high-dimensional output image, leveraging autograd. We further resolve potential intractability in PCA by spatially summing the Jacobians, reducing the feature space to a low-dimensional matrix regardless of spatial resolution. While the jacobian and calibration is fast (Table 8 and Figure 11), we recommend precomputing the jacobians for all data before calibration and testing.

*LLM Disclosure.* During the preparation of this manuscript, we used LLMs (Google Gemini) to polish the initial draft (grammar and style) and improve readability.

ACKNOWLEDGEMENTS

MC would like to acknowledge support from a fellowship from the Gulf Coast Consortia on the NLM Training Program in Biomedical Informatics and Data Science T15LM007093.

ETHICS STATEMENT

We develop uncertainty quantification methods for medical image segmentation metrics using publicly available and anonymized datasets. The methods are intended as research tools to improve reliability and interpretability, but are not validated for clinical deployment. Risks include potential misinterpretation of uncertainty estimates. We therefore release code for research purposes only and encourage further evaluation with clinical experts before translation to practice.

REPRODUCIBILITY STATEMENT

We describe the algorithms in our paper extensively in Appendix C. We also describe the experimental details in Appendix E. Our code is available at `https://github.com/matthewyccheung/compass`. All packages used in the repository are publicly available. All datasets used are publicly accessible.

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

## A  PROOF OF THEOREM 1

**Lemma 1** (Guaranteed Nestedness and Valid Scores). *For any $x \in \mathcal{X}$, let $m_x(b)$ be defined as in Theorem 1, and let*

$$S_\beta(x) := \left[ \min_{b \in [-\beta, +\beta]} m_x(b), \max_{b \in [-\beta, +\beta]} m_x(b) \right].$$

*This family $\{S_\beta(x)\}_{\beta \geq 0}$ is nested. Furthermore, the nonconformity score*

$$R(x, y) := \inf\{\beta \geq 0 : y \in S_\beta(x)\}$$

*is well-defined in $[0, \infty]$ and satisfies*

$$y \in S_\beta(x) \iff R(x, y) \leq \beta.$$

*Proof.* First, we prove nestedness. Let $0 \leq \beta_1 \leq \beta_2$. By definition, the perturbation interval $[-\beta_1, +\beta_1]$ is a subset of $[-\beta_2, +\beta_2]$. Let

$$S_{\beta_1}(x) = \left[ \min_{b \in [-\beta_1, +\beta_1]} m_x(b), \max_{b \in [-\beta_1, +\beta_1]} m_x(b) \right]$$

and

$$S_{\beta_2}(x) = \left[ \min_{b \in [-\beta_2, +\beta_2]} m_x(b), \max_{b \in [-\beta_2, +\beta_2]} m_x(b) \right].$$

The minimum of a function over a set is always greater than or equal to the minimum over a superset. Thus,

$$\min_{b \in [-\beta_1, +\beta_1]} m_x(b) \geq \min_{b \in [-\beta_2, +\beta_2]} m_x(b).$$

Conversely, the maximum over a set is less than or equal to the maximum over a superset. Thus,

$$\max_{b \in [-\beta_1, +\beta_1]} m_x(b) \leq \max_{b \in [-\beta_2, +\beta_2]} m_x(b).$$

Therefore, by definition of an interval, $S_{\beta_1}(x) \subseteq S_{\beta_2}(x)$, and the family is nested.

Second, we prove the equivalence $y \in S_\beta(x) \iff R(x, y) \leq \beta$. Since we have just proven that the sets $S_\beta(x)$ are nested, they expand monotonically in $\beta$. If $y \in S_{\beta'}(x)$ for some $\beta'$, then $y \in S_\beta(x)$ for all $\beta \geq \beta'$. The set $\{\beta \geq 0 : y \in S_\beta(x)\}$ is therefore a closed interval of the form $[R_0, \infty)$ (or $[0, \infty)$ if $y \in S_0(x)$). The infimum $R(x, y)$ is thus well-defined (as $R_0$), and $y \in S_\beta(x)$ if and only if $\beta \geq R(x, y)$. $\square$

**Lemma 2** (Exchangeability yields uniform ranks). *Let $(X_i, Y_i)_{i=1}^{n+1}$ be exchangeable, and let $R_i := R(X_i, Y_i)$ be derived from $D_{\mathrm{tr}}$. Then the rank of $R_{n+1}$ among $\{R_1, \ldots, R_n, R_{n+1}\}$ is uniformly distributed on $\{1, \ldots, n+1\}$, conditional on $D_{\mathrm{tr}}$.*

*Proof.* Since the nonconformity score $R(\cdot, \cdot)$ is a measurable, deterministic function of $(X_i, Y_i)$ and $D_{\mathrm{tr}}$, the exchangeability of the pairs $(X_i, Y_i)$ implies the exchangeability of the scalar scores $(R_i)_{i=1}^{n+1}$. Therefore, by the standard properties of exchangeable random variables, the rank of $R_{n+1}$ among all $n + 1$ scores is uniformly distributed on $\{1, \ldots, n+1\}$. $\square$

**Theorem 2** (Split-conformal coverage). *Under the assumptions of Theorem 1, for any $\alpha \in (0, 1)$, the prediction set $S_{\hat{\beta}}(X_{n+1})$ satisfies*

$$\mathbb{P}\left( Y_{n+1} \in S_{\hat{\beta}}(X_{n+1}) \,\middle|\, D_{\mathrm{tr}} \right) \geq 1 - \alpha.$$

*Proof.* By Lemma 1, our construction of $S_\beta(x)$ guarantees nestedness and also guarantees that $Y_{n+1} \in S_\beta(X_{n+1})$ if and only if $R_{n+1} \leq \beta$. Applying this to our calibrated quantile $\hat{\beta}$, we have

$$Y_{n+1} \in S_{\hat{\beta}}(X_{n+1}) \iff R_{n+1} \leq \hat{\beta}.$$

By Lemma 2, the rank of $R_{n+1}$ is uniform on $\{1, \ldots, n+1\}$. By definition, $\hat{\beta}$ is the $\lceil (1-\alpha)(n+1) \rceil$-th smallest value among the calibration scores $\{R_1, \ldots, R_n\}$. By the standard p-value argument for

split conformal prediction, the probability that the test score $R_{n+1}$ is less than or equal to $\hat{\beta}$ is bounded.

$$\mathbb{P}(R_{n+1} \leq \hat{\beta} \mid D_{\mathrm{tr}}) \geq 1 - \alpha.$$

Combining these, we have

$$\mathbb{P}\Big(Y_{n+1} \in S_{\hat{\beta}}(X_{n+1}) \mid D_{\mathrm{tr}}\Big) = \mathbb{P}(R_{n+1} \leq \hat{\beta} \mid D_{\mathrm{tr}}) \geq 1 - \alpha,$$

which proves the theorem. □

**Connection to Standard CP.** Theorem 1 generalizes the standard split conformal method. In the classical setting for regression, $S_{\beta}(x) = [\hat{y}(x) - \beta, \hat{y}(x) + \beta]$ corresponds to symmetric absolute-error balls around a predictor $\hat{y}(x)$. This family is trivially nested. Here, the sets $S_{\beta}(x)$ are generated by perturbations of latent features through $\Delta$ and evaluation under the downstream metric $h \circ g$. By rigorously defining $S_{\beta}(x)$ as the $[\min, \max]$ envelope, we guarantee the nestedness property required for the proof. The same rank-uniformity argument from exchangeability then ensures coverage. Thus, standard split conformal prediction is recovered as a special case of this framework.

## B  ASYMMETRIC PERTURBATIONS

We extend the symmetric method to provide more flexible and efficient prediction intervals to the asymmetric case. This approach is particularly valuable when the relationship between latent perturbations and the downstream metric is non-linear or asymmetric. For instance, a positive perturbation of a certain magnitude might cause a large increase in metric, while a negative perturbation of the same magnitude results in only a small decrease. It also allows for independent control over the rate of under- and over-estimation errors, specified by $\alpha_{\mathrm{lo}}$ and $\alpha_{\mathrm{hi}}$. To account for this, we calculate two distinct non-conformity scores ($R_{\mathrm{lo}}$, $R_{\mathrm{hi}}$) and two corresponding perturbation magnitudes ($\hat{\beta}_{\mathrm{lo}}$, $\hat{\beta}_{\mathrm{hi}}$). By calibrating the upper and lower bounds independently, Asymmetric COMPASS constructs an interval that adapts to these skewed error distributions, often resulting in tighter bounds than its symmetric counterpart in such cases.

**Theorem 3** (Asymmetric Split-Conformal Coverage). *Let the setup be the same as in Theorem 1, with metric function $m_x(b) := (h \circ g)(\hat{z}(x) + b\Delta)$. We define the one-sided envelope functions:*

$$L_x(\beta) := \min_{b \in [-\beta, 0]} m_x(b)$$
$$U_x(\beta) := \max_{b \in [0, +\beta]} m_x(b)$$

*These functions $L_x(\beta)$ and $U_x(\beta)$ are guaranteed to be non-increasing and non-decreasing in $\beta$, respectively.*

*For each calibration pair $(X_i, Y_i)$, define two non-conformity scores:*

$$R_{i,\mathrm{lo}} := \inf\{\beta \geq 0 : L_{X_i}(\beta) \leq Y_i\},$$
$$R_{i,\mathrm{hi}} := \inf\{\beta \geq 0 : U_{X_i}(\beta) \geq Y_i\}.$$

*Let $\alpha_{\mathrm{lo}}, \alpha_{\mathrm{hi}} \in (0,1)$ be user-specified miscoverage rates. Let $\hat{\beta}_{\mathrm{lo}}$ be the $\lceil(1 - \alpha_{\mathrm{lo}})(n + 1)\rceil$-th smallest value among $\{R_{1,\mathrm{lo}}, \ldots, R_{n,\mathrm{lo}}\}$, and let $\hat{\beta}_{\mathrm{hi}}$ be the $\lceil(1 - \alpha_{\mathrm{hi}})(n + 1)\rceil$-th smallest value among $\{R_{1,\mathrm{hi}}, \ldots, R_{n,\mathrm{hi}}\}$.*

*Then, for a fresh test pair $(X_{n+1}, Y_{n+1})$, the prediction set*

$$S(X_{n+1}) := [L_{X_{n+1}}(\hat{\beta}_{\mathrm{lo}}), \ U_{X_{n+1}}(\hat{\beta}_{\mathrm{hi}})]$$

*satisfies the marginal coverage guarantee*

$$\mathbb{P}\big(Y_{n+1} \in S(X_{n+1}) \mid D_{\mathrm{tr}}\big) \geq 1 - (\alpha_{\mathrm{lo}} + \alpha_{\mathrm{hi}}).$$

B.1 PROOF OF THEOREM 3

**Lemma 3** (Monotonic Bounds and Valid One-Sided Scores). *For any $x \in \mathcal{X}$, let $L_x(\beta)$ and $U_x(\beta)$ be defined as in Theorem 3. The function $L_x(\beta)$ is non-increasing in $\beta$, and the function $U_x(\beta)$ is non-decreasing in $\beta$. Furthermore, the nonconformity scores $R_{\mathrm{lo}}(x,y)$ and $R_{\mathrm{hi}}(x,y)$ are well-defined in $[0,\infty]$ and satisfy*

$$L_x(\beta) \leq y \quad \Longleftrightarrow \quad R_{\mathrm{lo}}(x,y) \leq \beta,$$
$$U_x(\beta) \geq y \quad \Longleftrightarrow \quad R_{\mathrm{hi}}(x,y) \leq \beta.$$

*Proof.* Let $0 \leq \beta_1 \leq \beta_2$. By definition, the interval $[-\beta_1, 0]$ is a subset of $[-\beta_2, 0]$. The minimum of a function over a set is always greater than or equal to the minimum over a superset. Thus:

$$L_x(\beta_1) = \min_{b \in [-\beta_1, 0]} m_x(b) \geq \min_{b \in [-\beta_2, 0]} m_x(b) = L_x(\beta_2).$$

This proves $L_x(\beta)$ is non-increasing. Similarly, $[0, +\beta_1]$ is a subset of $[0, +\beta_2]$. The maximum of a function over a set is always less than or equal to the maximum over a superset. Thus:

$$U_x(\beta_1) = \max_{b \in [0, +\beta_1]} m_x(b) \leq \max_{b \in [0, +\beta_2]} m_x(b) = U_x(\beta_2).$$

This proves $U_x(\beta)$ is non-decreasing.

This one-sided monotonicity ensures that if a bound is satisfied for some $\beta'$ (e.g., $L_x(\beta') \leq y$), it remains satisfied for all $\beta \geq \beta'$. Hence, the infima in the definitions of the scores exist, and the equivalences hold. $\qquad\square$

**Lemma 4** (Exchangeability yields uniform ranks). *Let $(X_i, Y_i)_{i=1}^{n+1}$ be exchangeable. Let $R_{i,\mathrm{lo}} := R_{\mathrm{lo}}(X_i, Y_i)$ and $R_{i,\mathrm{hi}} := R_{\mathrm{hi}}(X_i, Y_i)$ be derived from $D_{\mathrm{tr}}$. Then the ranks of $R_{n+1,\mathrm{lo}}$ among $\{R_{1,\mathrm{lo}}, \ldots, R_{n+1,\mathrm{lo}}\}$ and $R_{n+1,\mathrm{hi}}$ among $\{R_{1,\mathrm{hi}}, \ldots, R_{n+1,\mathrm{hi}}\}$ are each uniformly distributed on $\{1, \ldots, n+1\}$, conditional on $D_{\mathrm{tr}}$.*

*Proof.* Since the nonconformity scores are measurable, deterministic functions of $(X_i, Y_i)$ and $D_{\mathrm{tr}}$, the exchangeability of the data pairs implies the exchangeability of each sequence of scores, $(R_{i,\mathrm{lo}})_{i=1}^{n+1}$ and $(R_{i,\mathrm{hi}})_{i=1}^{n+1}$. Therefore, the rank of the test score within each sequence is uniform. $\qquad\square$

*Proof of Theorem 3.* The total miscoverage event is the union of two one-sided miscoverage events:

$$\mathbb{P}(Y_{n+1} \notin S(X_{n+1})) = \mathbb{P}(Y_{n+1} < L_{X_{n+1}}(\hat{\beta}_{\mathrm{lo}}) \text{ or } Y_{n+1} > U_{X_{n+1}}(\hat{\beta}_{\mathrm{hi}})).$$

By the union bound,

$$\mathbb{P}(Y_{n+1} \notin S(X_{n+1})) \leq \mathbb{P}(Y_{n+1} < L_{X_{n+1}}(\hat{\beta}_{\mathrm{lo}})) + \mathbb{P}(Y_{n+1} > U_{X_{n+1}}(\hat{\beta}_{\mathrm{hi}})).$$

By Lemma 3, these one-sided events are equivalent to events concerning the scores:

$$\mathbb{P}(Y_{n+1} \notin S(X_{n+1})) \leq \mathbb{P}(R_{n+1,\mathrm{lo}} > \hat{\beta}_{\mathrm{lo}}) + \mathbb{P}(R_{n+1,\mathrm{hi}} > \hat{\beta}_{\mathrm{hi}}).$$

By Lemma 4 and the construction of $\hat{\beta}_{\mathrm{lo}}$ and $\hat{\beta}_{\mathrm{hi}}$ as quantiles, the probabilities of these rank-based events are bounded by $\alpha_{\mathrm{lo}}$ and $\alpha_{\mathrm{hi}}$, respectively. Thus,

$$\mathbb{P}(Y_{n+1} \notin S(X_{n+1})) \leq \alpha_{\mathrm{lo}} + \alpha_{\mathrm{hi}}.$$

The claim for the coverage probability, $\mathbb{P}(Y_{n+1} \in S(X_{n+1})) \geq 1 - (\alpha_{\mathrm{lo}} + \alpha_{\mathrm{hi}})$, follows. $\qquad\square$

This theorem provides a rigorous guarantee for asymmetric intervals. The proof relies on the same core principles of exchangeability, but instead of assuming nestedness, we guarantee the required one-sided monotonicity by defining the bounds $L_x(\beta)$ and $U_x(\beta)$ as the $[\min, \max]$ over the respective perturbation ranges.

## C  ALGORITHMS

In the COMPASS framework, we decompose the model into an encoder $f$ that maps an input image $x$ to a representation $\hat{z}$, and a decoder $g$ that maps $\hat{z}$ to the final segmentation. A downstream metric function $h$ then maps the segmentation to a scalar metric $\hat{y}$. For simplicity, we show the algorithm using the endpoint (and not the conservative envelope that requires a perturbation sweep during the calibration procedure) – i.e., we define the perturbation and metric evaluation function $P(\hat{z}, \Delta, \beta)$ as the interval $[h(g(\hat{z} - \beta\Delta)), h(g(\hat{z} + \beta\Delta))]$. This endpoint construction is computationally efficient and theoretically valid given the monotonic metric response observed in our experiments (Figures 2 and 12). The conservative envelope variant follows the same approach, but the symmetric binary search involves a perturbation sweep for each sample during the calibration step to find the minimum and maximum.

For **COMPASS-L**, the representation $\hat{z}$ is the logits; the encoder $f$ is the entire network, and the decoder $g$ is the identity function. The perturbation direction is simply $\Delta = \mathbf{1}$.

For **COMPASS-J**, the representation $\hat{z}$ is an internal feature map. To find the sensitive direction $\Delta$, we first compute the Jacobian of the metric with respect to the features, $J = \nabla_{\hat{z}} h(g(\hat{z}))$. We then project this Jacobian onto the subspace spanned by the top $L$ principal components ($V_L$) of the training set Jacobians and reconstruct it. The final direction $\Delta$ is the normalized reconstruction: $\mathbf{d} = V_L V_L^\top J, \Delta = \frac{\mathbf{d}}{\|\mathbf{d}\|_2}$.

In general, the COMPASS framework follows several key steps:

1. **Training:** Assume an already trained base segmentation model, decomposed into an encoder $f$ and decoder $g$. If using **COMPASS-J**, compute the Jacobians of the metric with respect to the latent features $\hat{z}$ for the training set and perform PCA to identify the dominant sensitivity subspace $V_L$.

2. **Monotonicity Verification:** Perform a perturbation sweep on a subset of the data to verify that the metric response $m_x(b)$ is monotonic with respect to the perturbation magnitude along the computed direction $\Delta$. This empirical check justifies the use of the endpoints over the computationally expensive envelope construction.

3. **Calibration:** For each sample in the calibration set, compute the non-conformity score $R_i$ using a binary search. $R_i$ is the minimal perturbation magnitude $\beta$ such that the computed interval covers the ground truth. Finally, compute the calibrated quantile $\hat{\beta}$ from these scores using the standard finite-sample correction.

4. **Testing:** For a new test sample $x_{n+1}$, compute its specific perturbation direction $\Delta_{n+1}$ (using the learned subspace $V_L$ for COMPASS-J or $\mathbf{1}$ for COMPASS-L). Construct the final prediction interval using the calibrated $\hat{\beta}$ and the interval endpoints: $S_{\hat{\beta}}(x_{n+1}) = [m_{n+1}(-\hat{\beta}), m_{n+1}(+\hat{\beta})]$.

### C.1  TRAINING

---

**Algorithm 1** COMPASS Training

---

**Require:** Training data $D_{\mathrm{tr}}$, $L$ (number of components for COMPASS-J).
**Ensure:** Trained model $(f, g)$, and subspace $V_L$ (for COMPASS-J).

1:  **procedure** TRAINING($D_{\mathrm{tr}}$, $L$)
2:      Train the segmentation model $(f, g) \leftarrow \arg\min_{f,g} \mathcal{L}_{\mathrm{seg}}(D_{\mathrm{tr}})$.
3:      **if** using COMPASS-J **then**
4:          Compute Jacobians: $\mathcal{J} \leftarrow \{\sum_{\mathrm{spatial}} \nabla_{\hat{z}_j} h(g(f(x_j)))\}_{j \in D_{\mathrm{tr}}}$.
5:          Compute PCA: $V_L \leftarrow \mathrm{PCA}(\mathcal{J}, L)$.
6:          **return** $f, g, V_L$.
7:      **else**                                        ▷ COMPASS-L requires no extra training
8:          **return** $f, g$.
9:      **end if**
10: **end procedure**

---

## C.2 MONOTONICITY VERIFICATION

---

**Algorithm 2** Metric Monotonicity Verification

---

**Require:** Model $(f, g, V_L)$, Dataset $D = \{(x_i, y_i)\}_{i=1}^n$, $\beta_{\max}$, $N_{\text{steps}}$.
**Ensure:** Percentage of samples with monotonic metric response.

1: **procedure** VERIFY-MONOTONICITY($D$, model, $h$, $\beta_{\max}$, $N_{\text{steps}}$)
2:     **Initialize** $count \leftarrow 0$.
3:     **for** $i = 1, \ldots, n$ **do**
4:         $\hat{z}_i \leftarrow f(x_i)$
5:         **if** using COMPASS-J **then**
6:             $\mathbf{d}_i \leftarrow V_L V_L^\top \nabla_{\hat{z}} h(g(\hat{z}_i))$               ▷ Project Jacobian
7:             $\Delta_i \leftarrow \mathbf{d}_i / \|\mathbf{d}_i\|_2$                       ▷ Normalize
8:         **else**
9:             $\Delta_i \leftarrow \mathbf{1}$
10:         **end if**
11:         $B \leftarrow$ LinSpace($-\beta_{\max}, \beta_{\max}, N_{\text{steps}}$).        ▷ Discretized grid
12:         $M \leftarrow \emptyset$.
13:         **for** $b \in B$ **do**
14:             $M \leftarrow M \cup \{h(g(\hat{z}_i + b\Delta_i))\}$.        ▷ Evaluate metric
15:         **end for**
16:         **if** IS-SORTED-ASCENDING($M$) **then**     ▷ Check monotonicity
17:             $count \leftarrow count + 1$.
18:         **end if**
19:     **end for**
20:     **return** $count/n$
21: **end procedure**

---

Algorithm 2 is a diagnostic step performed after training. It verifies that perturbing the features along the computed direction $\Delta$ results in a monotonic change in the metric (area). High monotonicity validates the use of the efficient endpoint-based interval construction in the subsequent calibration and inference steps.

## C.3 SYMMETRIC COMPASS

---

**Algorithm 3** Symmetric COMPASS (Calibration & Inference)

---

**Require:** Calibration $D_{\text{cal}}$, Test $x_{n+1}$, $\alpha$, Model $(f, g, V_L)$.
**Ensure:** Prediction interval $S_{\hat{\beta}}(x_{n+1})$.

1: **procedure** CALIBRATION($D_{\text{cal}}$, model, $h$, $\alpha$)
2:     **Initialize** scores $\mathcal{R} \leftarrow \emptyset$.
3:     **for** $i = 1, \ldots, n$ **do**
4:         Define $\hat{z}_i$ and compute $\Delta_i$ (as in Alg. 2).
5:         $R_i \leftarrow$ SYMMETRIC-BINARY-SEARCH($\hat{z}_i, \Delta_i, y_i$).
6:         $\mathcal{R} \leftarrow \mathcal{R} \cup \{R_i\}$.
7:     **end for**
8:     $\hat{\beta} \leftarrow$ Quantile($\mathcal{R}, \frac{\lceil (1-\alpha)(n+1) \rceil}{n}$).
9:     **return** $\hat{\beta}$
10: **end procedure**

11: **procedure** INFERENCE($x_{n+1}$, model, $h$, $\hat{\beta}$)
12:     Define $\hat{z}_{n+1}$ and compute $\Delta_{n+1}$ (as in Alg. 2).
13:     $S_{\hat{\beta}}(x_{n+1}) \leftarrow P(\hat{z}_{n+1}, \Delta_{n+1}, \hat{\beta})$.
14:     **return** $S_{\hat{\beta}}(x_{n+1})$
15: **end procedure**

---

Algorithm 3 describes the symmetric formulation. During calibration, we compute a non-conformity score $R_i$ for each sample using a binary search (Algorithm 5). $R_i$ represents the minimal perturbation magnitude $\beta$ such that the interval $P(\hat{z}_i, \Delta_i, \beta)$ covers the true label $y_i$. We then compute the $(1 - \alpha)$ quantile of these scores, $\hat{\beta}$. During inference, we apply this calibrated $\hat{\beta}$ to the test sample to produce the final prediction interval.

## C.4 ASYMMETRIC COMPASS

---

**Algorithm 4** Asymmetric COMPASS (Calibration & Inference)

---

**Require:** Calibration $D_{\text{cal}}$, Test $x_{n+1}$, $\alpha_{\text{lo}}$, $\alpha_{\text{hi}}$, Model $(f, g, V_L)$.
**Ensure:** Prediction interval $S(x_{n+1})$.

1: **procedure** CALIBRATION($D_{\text{cal}}$, model, $h$, $\alpha_{\text{lo}}$, $\alpha_{\text{hi}}$)
2:     **Initialize** $\mathcal{R}_{\text{lo}} \leftarrow \emptyset, \mathcal{R}_{\text{hi}} \leftarrow \emptyset$.
3:     **for** $i = 1, \dots, n$ **do**
4:         Define $\hat{z}_i$ and compute $\Delta_i$ (as in Alg. 2).
5:         $(R_{i,\text{lo}}, R_{i,\text{hi}}) \leftarrow$ ASYMMETRIC-BINARY-SEARCH$(\hat{z}_i, \Delta_i, y_i)$.
6:         $\mathcal{R}_{\text{lo}} \leftarrow \mathcal{R}_{\text{lo}} \cup \{R_{i,\text{lo}}\}$.
7:         $\mathcal{R}_{\text{hi}} \leftarrow \mathcal{R}_{\text{hi}} \cup \{R_{i,\text{hi}}\}$.
8:     **end for**
9:     $\hat{\beta}_{\text{lo}} \leftarrow$ Quantile$(\mathcal{R}_{\text{lo}}, \frac{\lceil (1-\alpha_{\text{lo}})(n+1) \rceil}{n})$.
10:    $\hat{\beta}_{\text{hi}} \leftarrow$ Quantile$(\mathcal{R}_{\text{hi}}, \frac{\lceil (1-\alpha_{\text{hi}})(n+1) \rceil}{n})$.
11:    **return** $\hat{\beta}_{\text{lo}}, \hat{\beta}_{\text{hi}}$
12: **end procedure**

13: **procedure** INFERENCE($x_{n+1}$, model, $h$, $\hat{\beta}_{\text{lo}}$, $\hat{\beta}_{\text{hi}}$)
14:    Define $\hat{z}_{n+1}$ and compute $\Delta_{n+1}$ (as in Alg. 2).
15:    $S(x_{n+1}) \leftarrow [h(g(\hat{z}_{n+1} - \hat{\beta}_{\text{lo}}\Delta_{n+1})), h(g(\hat{z}_{n+1} + \hat{\beta}_{\text{hi}}\Delta_{n+1}))]$.
16:    **return** $S(x_{n+1})$
17: **end procedure**

---

Algorithm 4 extends the framework to the asymmetric case. We independently calibrate a lower bound ($\hat{\beta}_{\text{lo}}$) and an upper bound ($\hat{\beta}_{\text{hi}}$) using separate non-conformity scores. This allows the interval to expand differently in the positive and negative directions, which is efficient for metrics with asymmetric sensitivity.

## C.5 SCORE-FINDING PROCEDURES

---

**Algorithm 5** Symmetric Binary Search

---

1: **function** SYMMETRIC-BINARY-SEARCH$(\hat{z}, \Delta, y)$
2:     $b_{\text{low}} \leftarrow 0, b_{\text{high}} \leftarrow \beta_{\text{range}}$.
3:     **for** $k = 1, \dots, k_{\max}$ **do**
4:         $b_{\text{mid}} \leftarrow (b_{\text{low}} + b_{\text{high}})/2$.
5:         $S_{\text{mid}} \leftarrow P(\hat{z}, \Delta, b_{\text{mid}})$         ▷ Compute endpoint interval
6:         **if** $y \in S_{\text{mid}}$ **then** $b_{\text{high}} \leftarrow b_{\text{mid}}$
7:         **else** $b_{\text{low}} \leftarrow b_{\text{mid}}$
8:         **end if**
9:     **end for**
10:    **return** $b_{\text{high}}$.
11: **end function**

---

---

**Algorithm 6** Asymmetric Binary Search

---

1: **function** ASYMMETRIC-BINARY-SEARCH($\hat{z}, \Delta, y$)
2:     $m_{\text{lo}}(b) \leftarrow h(g(\hat{z} - b\Delta))$, $m_{\text{hi}}(b) \leftarrow h(g(\hat{z} + b\Delta))$
3:     *Find Upper Bound Score ($R_{\text{hi}}$):*
4:     $b_{\text{low}} \leftarrow 0, b_{\text{high}} \leftarrow \beta_{\text{range}}$.
5:     **for** $k = 1, \ldots, k_{\text{max}}$ **do**
6:         $b_{\text{mid}} \leftarrow (b_{\text{low}} + b_{\text{high}})/2$.
7:         **if** $y \leq m_{\text{hi}}(b_{\text{mid}})$ **then** $b_{\text{high}} \leftarrow b_{\text{mid}}$
8:         **else** $b_{\text{low}} \leftarrow b_{\text{mid}}$
9:         **end if**
10:     **end for**
11:     $R_{\text{hi}} \leftarrow b_{\text{high}}$.
12:     *Find Lower Bound Score ($R_{\text{lo}}$):*
13:     $b_{\text{low}} \leftarrow 0, b_{\text{high}} \leftarrow \beta_{\text{range}}$.
14:     **for** $k = 1, \ldots, k_{\text{max}}$ **do**
15:         $b_{\text{mid}} \leftarrow (b_{\text{low}} + b_{\text{high}})/2$.
16:         **if** $y \geq m_{\text{lo}}(b_{\text{mid}})$ **then** $b_{\text{high}} \leftarrow b_{\text{mid}}$
17:         **else** $b_{\text{low}} \leftarrow b_{\text{mid}}$
18:         **end if**
19:     **end for**
20:     $R_{\text{lo}} \leftarrow b_{\text{high}}$.
21:     **return** $(R_{\text{lo}}, R_{\text{hi}})$.
22: **end function**

---

These are the root-finding functions necessary to determine the non-conformity scores. We use a binary search for computational efficiency. This requires $\mathcal{O}(\log N)$ forward passes, as opposed to $\mathcal{O}(N)$ for a linear search, making it highly scalable.

# D ADDITIONAL TABLES

Table 2: **Empirical coverage for U-Net.** All methods reach (close to) target coverage.

| Dataset | $\alpha$ | Coverage (Mean±STD) | | | | | |
|---|---|---|---|---|---|---|---|
| | | COMPASS-J | COMPASS-L | E2E-CQR | Local | Output-CQR | SCP |
| H&E | 0.05 | 0.952±0.023 | 0.953±0.02 | 0.955±0.025 | 0.95±0.021 | 0.951±0.019 | 0.956±0.018 |
| H&E | 0.10 | 0.907±0.029 | 0.907±0.03 | 0.903±0.028 | 0.903±0.03 | 0.903±0.031 | 0.91±0.03 |
| H&E | 0.15 | 0.856±0.03 | 0.856±0.031 | 0.856±0.035 | 0.848±0.036 | 0.85±0.032 | 0.849±0.032 |
| Skin Lesion | 0.05 | 0.951±0.009 | 0.953±0.008 | 0.951±0.01 | 0.95±0.01 | 0.951±0.009 | 0.951±0.01 |
| Skin Lesion | 0.10 | 0.9±0.014 | 0.904±0.013 | 0.9±0.012 | 0.899±0.012 | 0.897±0.013 | 0.899±0.013 |
| Skin Lesion | 0.15 | 0.853±0.015 | 0.86±0.016 | 0.853±0.016 | 0.852±0.015 | 0.852±0.015 | 0.853±0.015 |
| Nodule | 0.05 | 0.949±0.018 | 0.953±0.018 | 0.958±0.014 | 0.948±0.017 | 0.954±0.016 | 0.953±0.019 |
| Nodule | 0.10 | 0.901±0.025 | 0.904±0.026 | 0.904±0.022 | 0.906±0.025 | 0.908±0.024 | 0.906±0.024 |
| Nodule | 0.15 | 0.852±0.032 | 0.858±0.031 | 0.849±0.03 | 0.85±0.028 | 0.845±0.026 | 0.857±0.027 |
| PolyP | 0.05 | 0.955±0.02 | 0.96±0.018 | 0.957±0.019 | 0.951±0.022 | 0.955±0.018 | 0.96±0.022 |
| PolyP | 0.10 | 0.899±0.034 | 0.908±0.032 | 0.902±0.027 | 0.897±0.032 | 0.901±0.029 | 0.907±0.03 |
| PolyP | 0.15 | 0.849±0.035 | 0.865±0.03 | 0.853±0.034 | 0.854±0.037 | 0.856±0.033 | 0.856±0.032 |

Table 3: **Comparison between COMPASS and FCP interval lengths.** Because FCP fails to reliably converge when finding a non-conformity score, we instead provide a comparison, where we empirically find the minimal radius of an $\mathcal{L}_p$ ball that achieves the target coverage across the test set. COMPASS methods consistently achieve the most efficient interval lengths.

| Dataset | $\alpha$ | Interval Size (Pixels$^2$, Mean±STD) | | |
|---|---|---|---|---|
| | | COMPASS-J | COMPASS-L | FCP |
| H&E | 0.05 | 4637±630 | **4408±432** | 9702±1535 |
| H&E | 0.10 | 3160±336 | **3139±375** | 6463±1193 |
| H&E | 0.15 | **2320±252** | 2354±146 | 4979±647 |
| Skin Lesion | 0.05 | **1657±80** | 1689±83 | 14838±10 |
| Skin Lesion | 0.10 | **1179±53** | 1208±58 | 12445±51 |
| Skin Lesion | 0.15 | **934±30** | 956±33 | 8267±79 |
| Nodule | 0.05 | **3257±210** | 3394±280 | 16247±2 |
| Nodule | 0.10 | **2444±174** | 2510±180 | 16143±3 |
| Nodule | 0.15 | **2016±143** | 2082±142 | 16056±4 |
| PolyP | 0.05 | **5489±575** | 6376±769 | 15937±9 |
| PolyP | 0.10 | **4056±293** | 4397±469 | 15644±10 |
| PolyP | 0.15 | **3394±290** | 3686±361 | 15353±14 |

Table 4: **Comparison of layer choice.** For $\alpha = 0.1$, we find that while for some datasets the efficiency gains are comparable, penultimate features generally achieve shorter interval lengths.

| Dataset | Interval Size (Pixels$^2$, Mean±STD) | | | |
|---|---|---|---|---|
| | COMPASS-J (Bottleneck) | COMPASS-J (Deep) | COMPASS-J (Shallow) | COMPASS-L |
| H&E | 9394±799 | 3160±336 | 3140±386 | **3139±375** |
| Skin Lesion | 4084±154 | **1179±53** | 1210±51 | 1208±58 |
| Nodule | 8070±494 | **2444±174** | 2500±187 | 2510±180 |
| PolyP | Non-monotonic | **4056±293** | 4222±390 | 4397±469 |

Table 5: **Comparison between baseline and COMPASS methods for Weighted CP.** For $\alpha = 0.1$, weighted COMPASS methods consistently outperform baseline methods in terms of interval length and restoring coverage under covariate shift.

| Dataset | Weighting | Method | Interval Size (Mean±STD, Pixels$^2$) | Coverage (Mean±STD) |
|---|---|---|---|---|
| H&E | Class | COMPASS-J | 2055±220 | 0.911±0.024 |
| | | COMPASS-L | **2065±263** | 0.903±0.024 |
| | | E2E-CQR | 2444±278 | 0.900±0.027 |
| | | Local | 3113±201 | 0.907±0.024 |
| | | Output-CQR | 2890±265 | 0.901±0.024 |
| | | SCP | 2371±290 | 0.900±0.026 |
| | Feature | COMPASS-J | **1916±177** | 0.909±0.024 |
| | | COMPASS-L | 1944±244 | 0.896±0.024 |
| | | E2E-CQR | 2197±288 | 0.884±0.029 |
| | Jacobian | COMPASS-J | **1822±200** | 0.896±0.031 |
| | Unweighted | COMPASS-J | 1810±127 | 0.886±0.025 |
| | | COMPASS-L | 1759±159 | 0.884±0.024 |
| | | E2E-CQR | 1993±196 | 0.871±0.026 |
| | | Local | 2833±158 | 0.88±0.027 |
| | | Output-CQR | 2563±134 | 0.879±0.024 |
| | | SCP | 2003±169 | 0.878±0.024 |
| Skin Lesion | Class | COMPASS-J | **1318±132** | 0.896±0.019 |
| | | COMPASS-L | 1373±136 | 0.895±0.019 |
| | | E2E-CQR | 1335±277 | 0.887±0.021 |
| | | Local | 2426±354 | 0.893±0.02 |
| | | Output-CQR | 3380±43 | 0.888±0.018 |
| | | SCP | 1905±475 | 0.888±0.024 |
| | Feature | COMPASS-J | **1573±158** | 0.919±0.014 |
| | | COMPASS-L | 1669±168 | 0.919±0.013 |
| | | E2E-CQR | 2030±269 | 0.919±0.012 |
| | Jacobian | COMPASS-J | **1485±126** | 0.914±0.013 |
| | Unweighted | COMPASS-J | **2183±132** | 0.94±0.005 |
| | | COMPASS-L | 2211±150 | 0.939±0.007 |
| | | E2E-CQR | 3400±226 | 0.945±0.006 |
| | | Local | 5446±380 | 0.947±0.006 |
| | | Output-CQR | 4201±145 | 0.945±0.008 |
| | | SCP | 4707±323 | 0.948±0.006 |

Table 6: **COMPASS produces the most efficient interval lengths on the majority of datasets across target coverages for SegResNet.**

| | | Interval Size (Pixels$^2$, Mean±STD) | | | | | |
|---|---|---|---|---|---|---|---|
| Dataset | $\alpha$ | COMPASS-J | COMPASS-L | E2E-CQR | Local | Output-CQR | SCP |
| H&E | 0.05 | 6129±1147 | **4680±595** | 5115±541 | 7649±821 | 6860±571 | 6021±858 |
| | 0.10 | 3283±262 | **3217±247** | 3575±344 | 4800±700 | 4509±389 | 3911±371 |
| | 0.15 | 2483±226 | **2400±244** | 2563±182 | 3491±333 | 3358±285 | 2783±317 |
| Skin Lesion | 0.05 | **1599±97** | 1829±191 | 2281±146 | 4034±256 | 10942±53 | 2987±227 |
| | 0.10 | **1108±45** | 1146±45 | 1236±66 | 2311±167 | 3056±45 | 1685±135 |
| | 0.15 | 873±27 | 927±27 | **857±52** | 1710±64 | 2293±32 | 1083±49 |
| Nodule | 0.05 | 3675±480 | 3577±413 | 4461±337 | **3513±212** | 6197±64 | 4068±388 |
| | 0.10 | 2671±251 | **2645±216** | 3390±197 | 2762±129 | 3942±102 | 2826±200 |
| | 0.15 | **2125±203** | 2127±203 | 2646±128 | 2355±131 | 2937±95 | 2188±167 |
| PolyP | 0.05 | **5890±417** | 6068±435 | 8495±1012 | 7992±1324 | 8296±822 | 7774±930 |
| | 0.10 | **4507±529** | 4737±486 | 4941±949 | 4837±611 | 5061±599 | 5169±559 |
| | 0.15 | 3188±500 | 3569±572 | **2668±255** | 3594±326 | 3556±401 | 3467±389 |

Table 7: **Empirical coverage for SegResNet.** All methods reach (close to) target coverage.

| Dataset | $\alpha$ | Coverage (Mean±STD) | | | | | |
|---------|----------|-----------|-----------|----------|-------|------------|-----|
| | | COMPASS-J | COMPASS-L | E2E-CQR | Local | Output-CQR | SCP |
| H&E | 0.05 | 0.956±0.019 | 0.949±0.021 | 0.954±0.021 | 0.949±0.022 | 0.949±0.023 | 0.953±0.02 |
| | 0.10 | 0.905±0.029 | 0.9±0.027 | 0.91±0.027 | 0.901±0.028 | 0.902±0.028 | 0.906±0.027 |
| | 0.15 | 0.858±0.033 | 0.852±0.034 | 0.856±0.031 | 0.854±0.033 | 0.854±0.033 | 0.856±0.035 |
| Skin Lesion | 0.05 | 0.952±0.011 | 0.95±0.01 | 0.952±0.01 | 0.95±0.011 | 0.951±0.01 | 0.952±0.01 |
| | 0.10 | 0.903±0.014 | 0.901±0.013 | 0.9±0.014 | 0.902±0.016 | 0.901±0.013 | 0.902±0.015 |
| | 0.15 | 0.854±0.015 | 0.853±0.016 | 0.851±0.017 | 0.852±0.015 | 0.854±0.014 | 0.854±0.015 |
| Nodule | 0.05 | 0.958±0.017 | 0.953±0.017 | 0.954±0.019 | 0.951±0.019 | 0.953±0.017 | 0.952±0.018 |
| | 0.10 | 0.902±0.026 | 0.901±0.027 | 0.909±0.025 | 0.9±0.024 | 0.901±0.027 | 0.904±0.026 |
| | 0.15 | 0.852±0.026 | 0.85±0.026 | 0.856±0.029 | 0.848±0.032 | 0.849±0.032 | 0.851±0.03 |
| PolyP | 0.05 | 0.954±0.022 | 0.948±0.022 | 0.954±0.022 | 0.949±0.023 | 0.95±0.021 | 0.952±0.02 |
| | 0.10 | 0.905±0.032 | 0.899±0.033 | 0.909±0.025 | 0.904±0.032 | 0.9±0.032 | 0.908±0.03 |
| | 0.15 | 0.858±0.035 | 0.85±0.035 | 0.856±0.031 | 0.856±0.031 | 0.855±0.037 | 0.859±0.033 |

## E EXPERIMENTAL DETAILS

**Architectures.** We use the standard U-Net, SegResNet, and SwinUNETR architecture available in the MONAI framework (Cardoso et al., 2022). For the U-Net, we used an encoder with channel sizes of (32, 64, 128, 256) and a corresponding decoder, using two residual units per block, batch normalization, and a dropout rate of 0.1. For SegResNet, we used a residual encoder-decoder network that started with 32 initial filters and included varying numbers of blocks in its down-sampling path (1, 2, 2, 4), batch normalization, and a dropout probability of 0.1. For SwinUNETR, we used a transformer-based model configured for an image size of (128, 128) pixels, with a feature size of 48 and multi-head attention mechanisms across four depth levels.

**Preprocessing.** To standardize all experiments, we resize each image and segmentation mask to a standard resolution of 128x128 pixels. For data augmentation during training, we apply a set of transformations consisting of a random crop to the specified size, followed by random horizontal and vertical flips, each with a equal probability.

**Training details.** For each architecture, we trained both a standard model and an End-to-End Conformalized Quantile Regression (E2E-CQR) variant (Lambert et al., 2024), with all models being trained using an AdamW optimizer with a learning rate of 1e-4 and a batch size of 32. The standard models were configured for binary segmentation tasks, taking a 1 (for grayscale) or 3-channel (for RGB image) as input and producing a single-channel output mask. Training was optimized using a Dice score loss function to maximize the overlap between the predicted and ground-truth segmentations. Following the original E2E-CQR model (Lambert et al., 2024), we modify the number of output channels to 3 to produce three distinct segmentation masks, corresponding to the lower quantile, the median prediction, and the upper quantile, which together form the prediction interval. These models were trained using the Tversky loss (Salehi et al., 2017). Thus, it allows the model to directly learn the uncertainty bounds end-to-end.

**Baseline Calibration Details.** We follow the standard frameworks for SCP (Papadopoulos et al., 2002; Lei et al., 2018), CQR (Romano et al., 2019), and Local CP (Papadopoulos et al., 2008; 2011; Lei et al., 2018). For CQR, we use two separate Gradient Boosting Regressors to learn the lower and upper quantiles of the prediction and adjust the resulting We use a learning rate of 0.1, 50 estimators, a maximum depth of 3, minimum leaf samples of 1, and a minimum samples to split of 9. For Local CP, we use two separate Random Forest Regressors to learn the mean and mean absolute difference. We use 1000 estimators and a minimum leaf sample of 100. For E2E-CQR, we follow the same calibration procedure as CQR. However, in this case, the quantiles are learned by the model.

**COMPASS Calibration and Testing Details.** For all experiments, we run both the symmetric and asymmetric versions of COMPASS with 1 component on the logits and each layer in the segmentation head. We report the layer with the minimum mean interval length and its corresponding

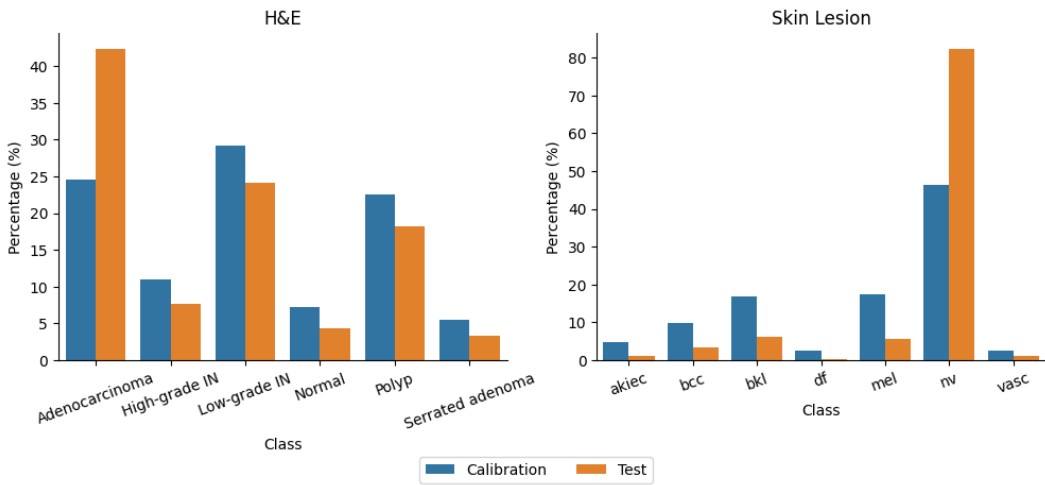

Figure 6: **Adversarial shift distribution for H&E and Skin Lesion datasets.** For H&E, we place 60% and 40% of the Adenocarcinoma (easy) samples into the calibration and test sets. For Skin Lesion, we place 70% and 30% of the melanocytic nevi (hard) samples into the calibration and test sets.

coverage. We perform 100 random calibration-test splits. For weighted CP, we use LightGBM with the default settings (Ke et al., 2017). We also perform 100 calibration-test splits. However, in this case, the samples were shuffled to maintain the target proportion of classes. We show the covariate shift in Figure 6.

# F    PRACTICAL RECOMMENDATION ADDITIONAL RESULTS

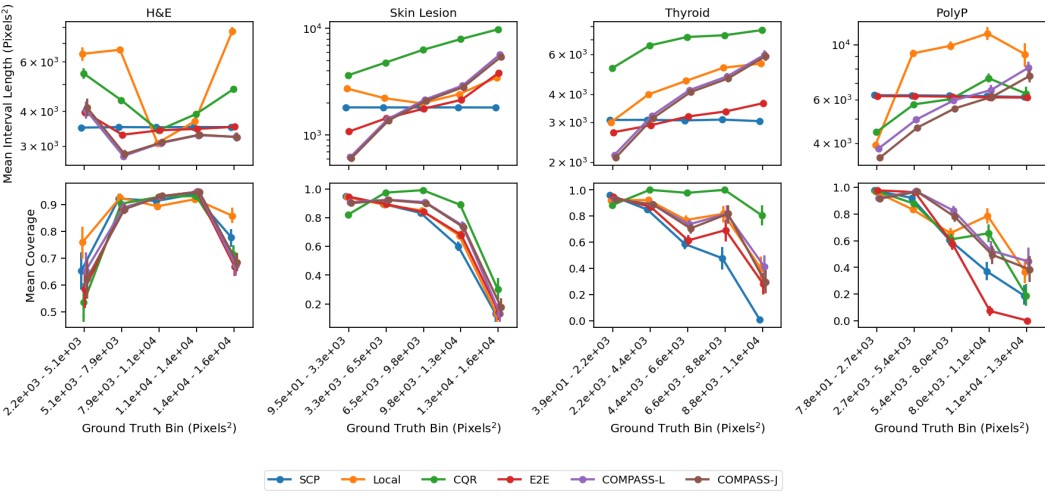

Figure 7: **COMPASS results in stable calibration.** We use the results from $\alpha$=0.1, binned the ground truth values into 5 non-overlapping bins, and computed the mean interval length and mean coverage (and their 95% confidence intervals). We find that COMPASS methods tend to perform better for shorter interval lengths (except for H&E), achieving approximately the same coverage. The count of the bins tends to be a better indicator of calibration stability (See Figure 8). Please note that our guarantees are only marginal; therefore, coverage may be lower for values that are less frequently represented during calibration.

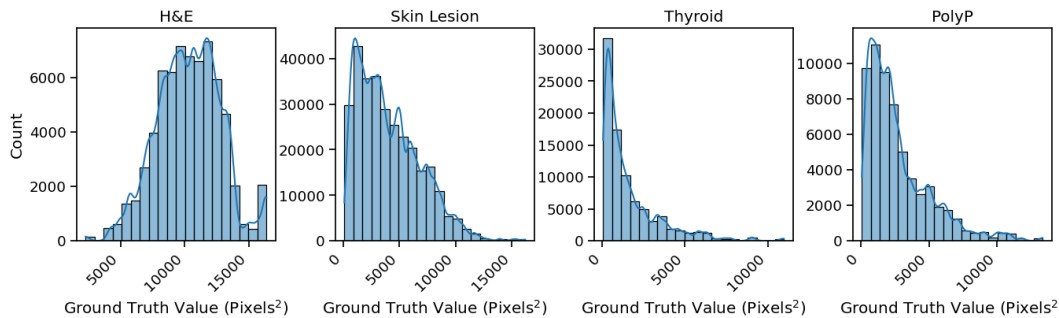

Figure 8: **Ground Truth Distribution.** We show histogram plots of the distribution of ground truth segmentation area values in Pixels$^2$ with a KDE overlaid.

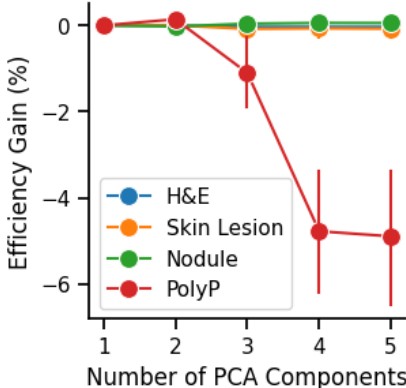

Figure 9: **One principal component is generally sufficient for COMPASS-J.** For 4 datasets, we run COMPASS-J with 1 to 5 components for 100 iterations. We find that there is no additional benefits of using more than 1 principal component, as the first principal component already explained most of the variance (Figure 5).

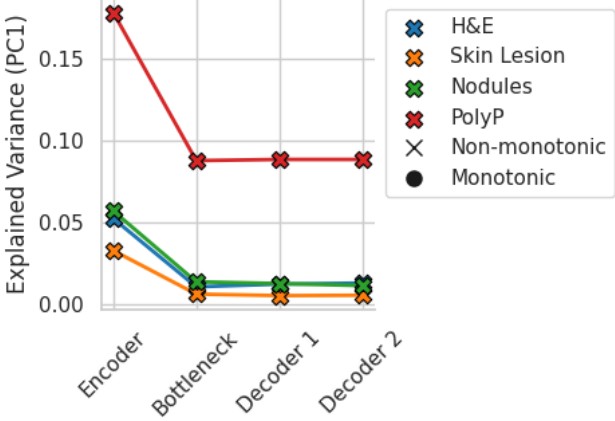

Figure 10: **Explained variance of the Jacobians is significantly lower when computed on raw flattened features.** For our 4 datasets, we plot the first principal component's explained variance against the feature layer used for COMPASS-J. We find all layers result in non-monotonicity, indicating the translation-variant nature of the flattened space.

Table 8: **Jacobian computation for COMPASS-J is fast.** We present the average computational time to compute the Jacobians of feature size 64x64x64 across four datasets below using a single NVIDIA A100 GPU. We compute the Jacobians with respect to the metric, which is extremely fast with autograd. This is because the downstream metric is one-dimensional. Furthermore, to expedite the training and calibration step, we precompute and save the Jacobians summed on the spatial dimension.

| | H&E | Skin Lesion | Nodule | PolyP |
|---|---|---|---|---|
| Compute time for full dataset (Mean±STD in seconds) | 73.6±0.6 | 305.8±1.1 | 93.3±0.9 | 73.6±0.6 |

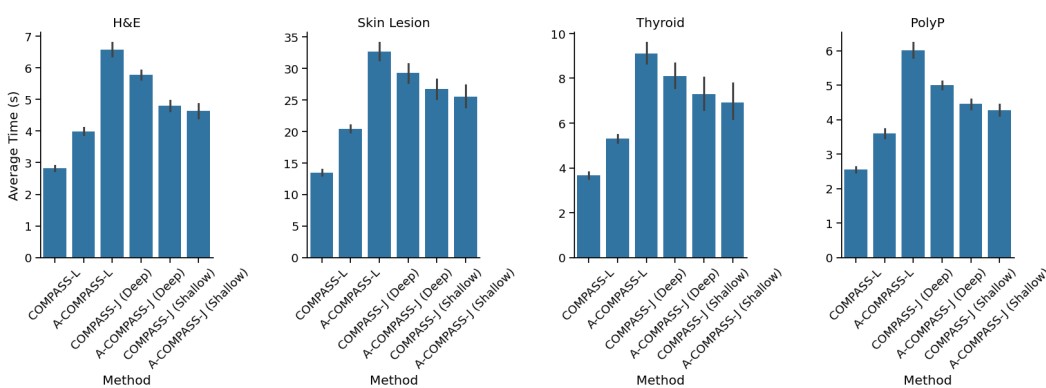

Figure 11: **COMPASS methods do not require substantial computing times.** The calibration step requires multiple forward passes. However, we find that combining forward passes and binary search enables fast calibration. We show the average computational time required for calibration (for 100 repeats) across 4 datasets using a single NVIDIA A100 GPU. We compute times for the symmetric (without A-) and asymmetric (A-), deep (-Deep) and (-Shallow) layers, and COMPASS-L versus COMPASS-J. The full calibration runtimes are on the order of seconds for the full calibration dataset. The number of samples used for calibration was 223 (H&E), 1000 (Skin Lesion), 349 (Thyroid), and 200 (PolyP).

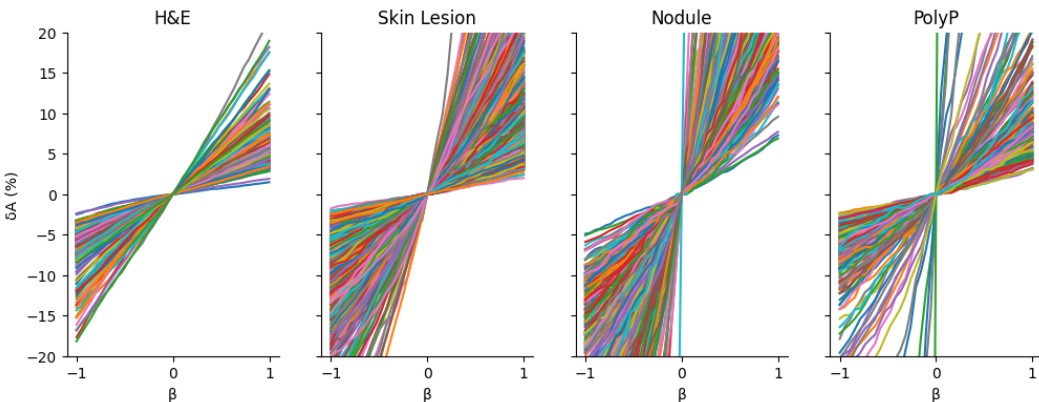

Figure 12: **Monotonicity verification.** We verify the nestedness of linear latent perturbations (Definition1). For each dataset, we perform a sweep across different $\beta$s and compute the change in volume $\delta A = \frac{A_\beta - A_0}{A_0} \times 100$ where $A_\beta$ is the area when the original latent is perturbed by $\beta$.

## G SEGMENTATION OVERLAYS FOR COMPASS-J AND COMPASS-L

### G.1 SEGMENTATION OVERLAYS FOR DIFFERENT PERCENTAGES OF CHANGE IN SEGMENTATION AREA ($\delta A$) FOR COMPASS-J

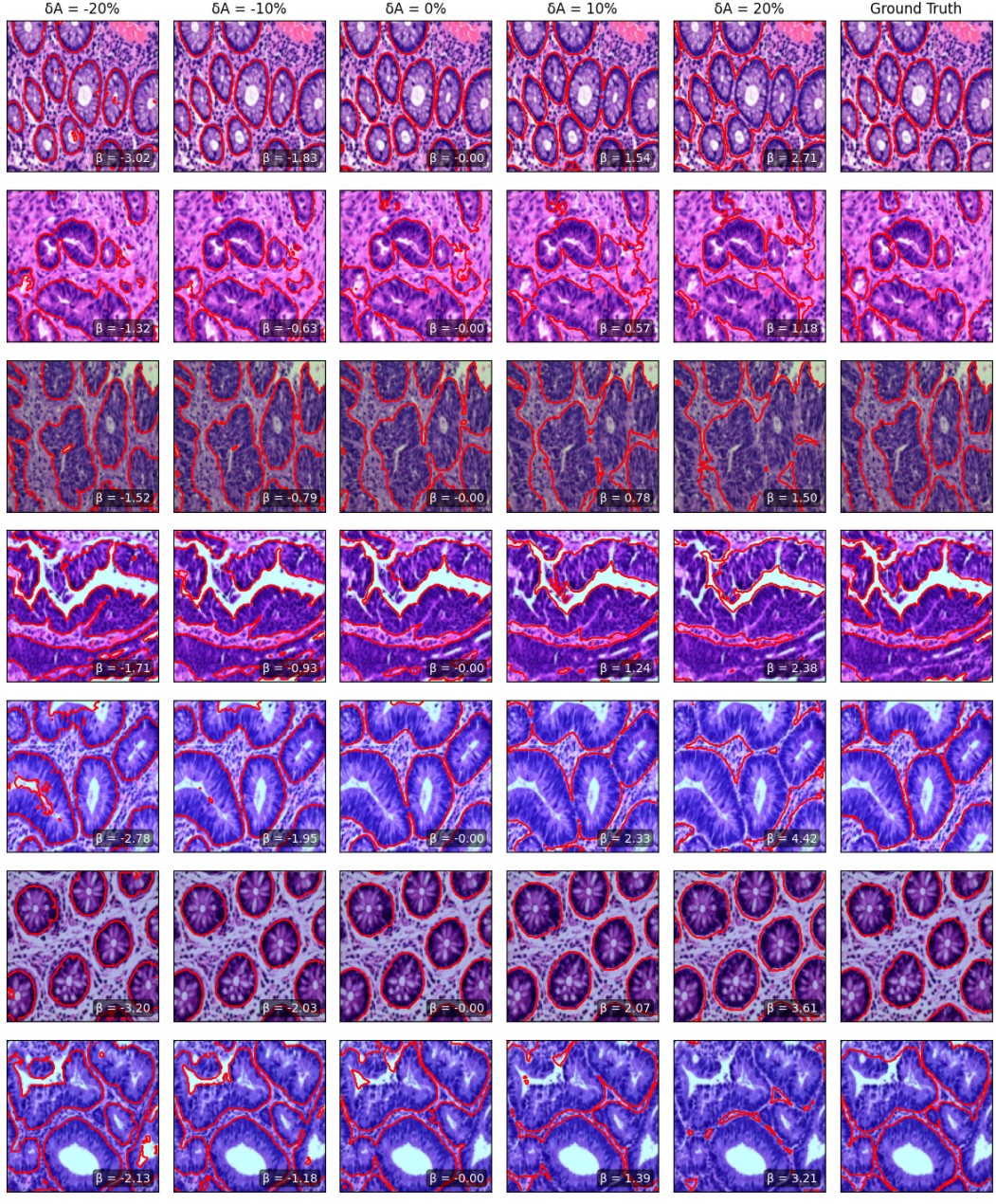

Figure 13: Segmentation area increases with $\beta$ for H&E. Each row is a different sample. $\delta A = 0\%$ is the original prediction.

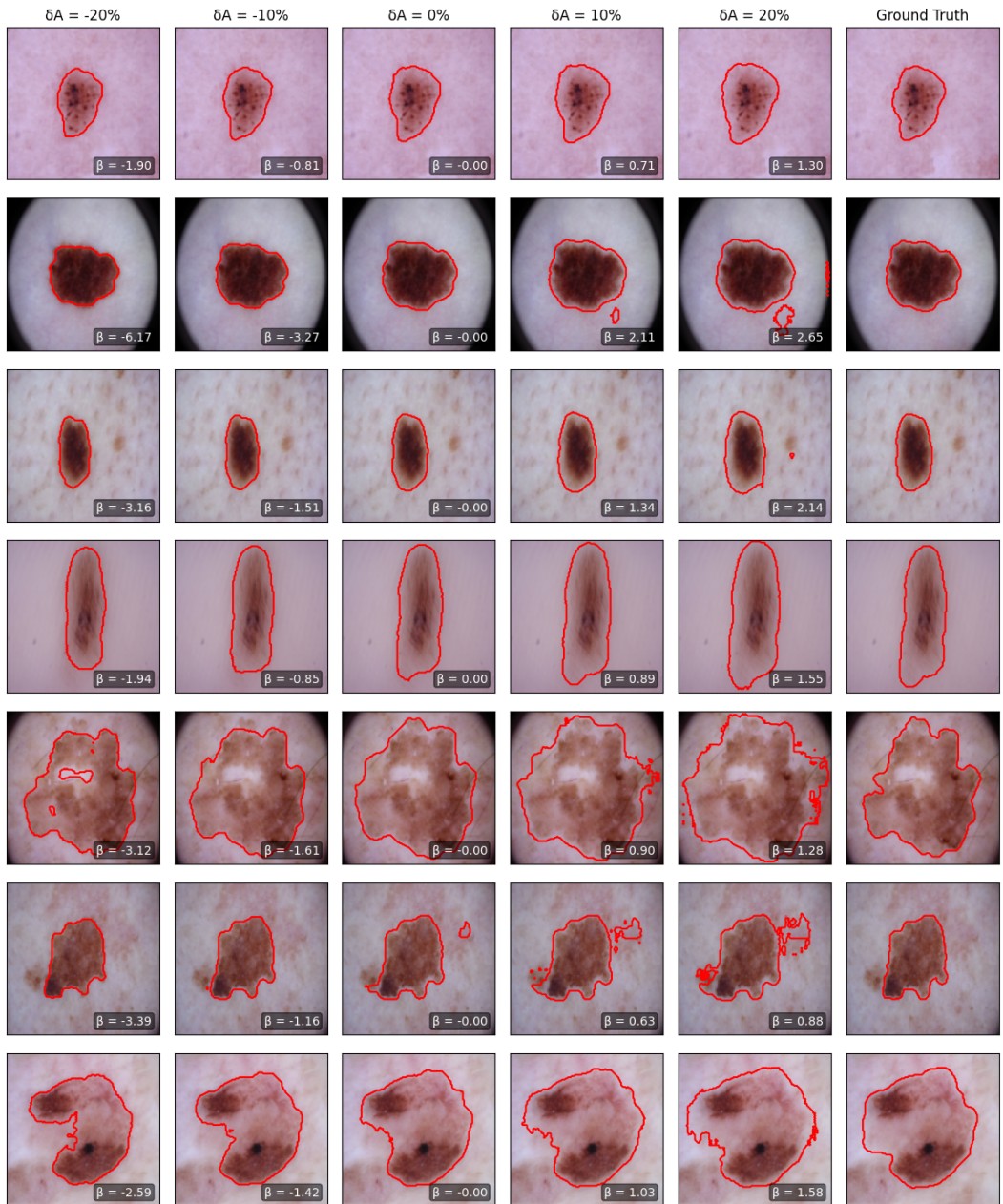

Figure 14: Segmentation area increases with $\beta$ for Skin Lesion. Each row is a different sample. $\delta A = 0\%$ is the original prediction.

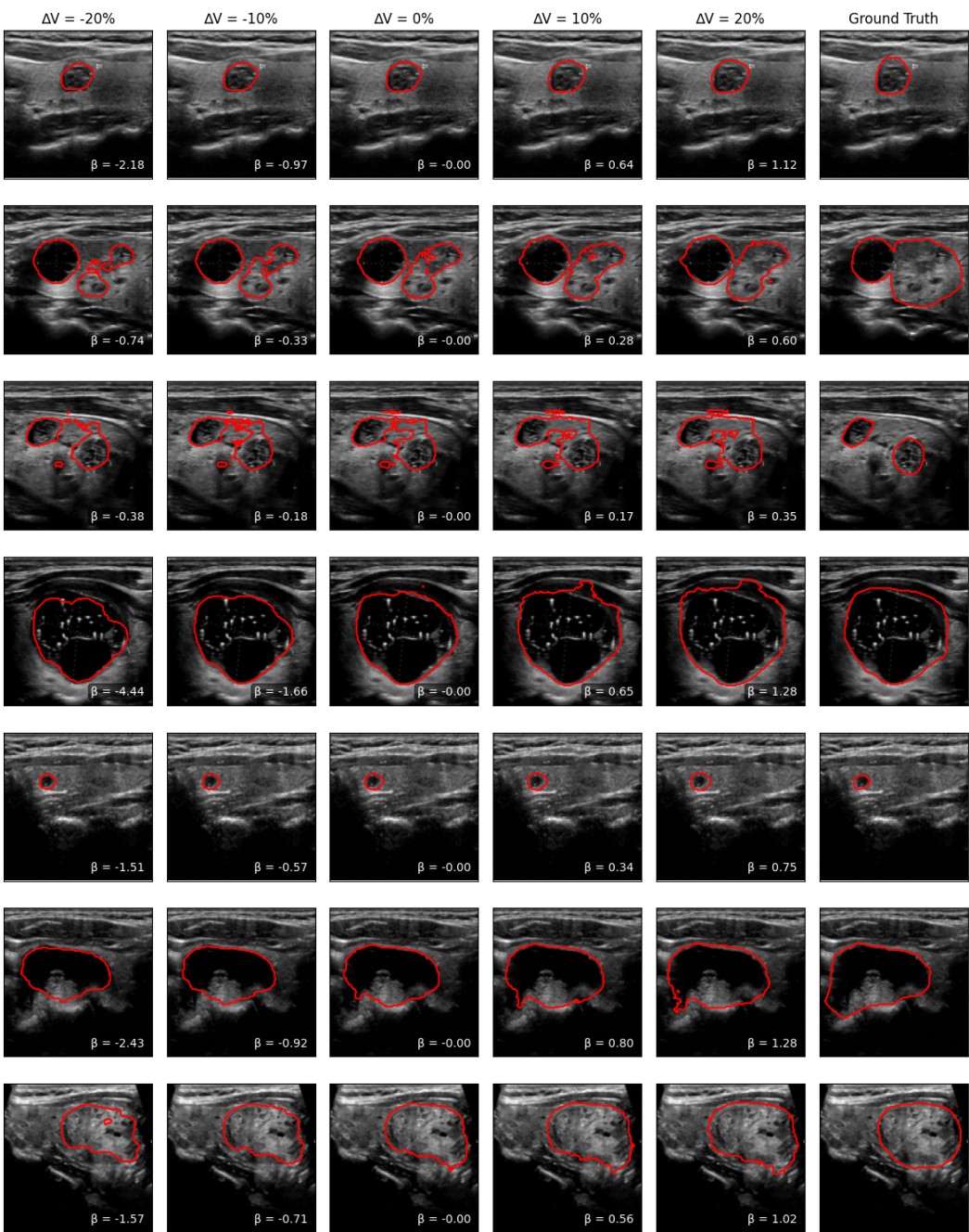

Figure 15: Segmentation area increases with $\beta$ for Thyroid Nodule. Each row is a different sample. $\delta A = 0\%$ is the original prediction.

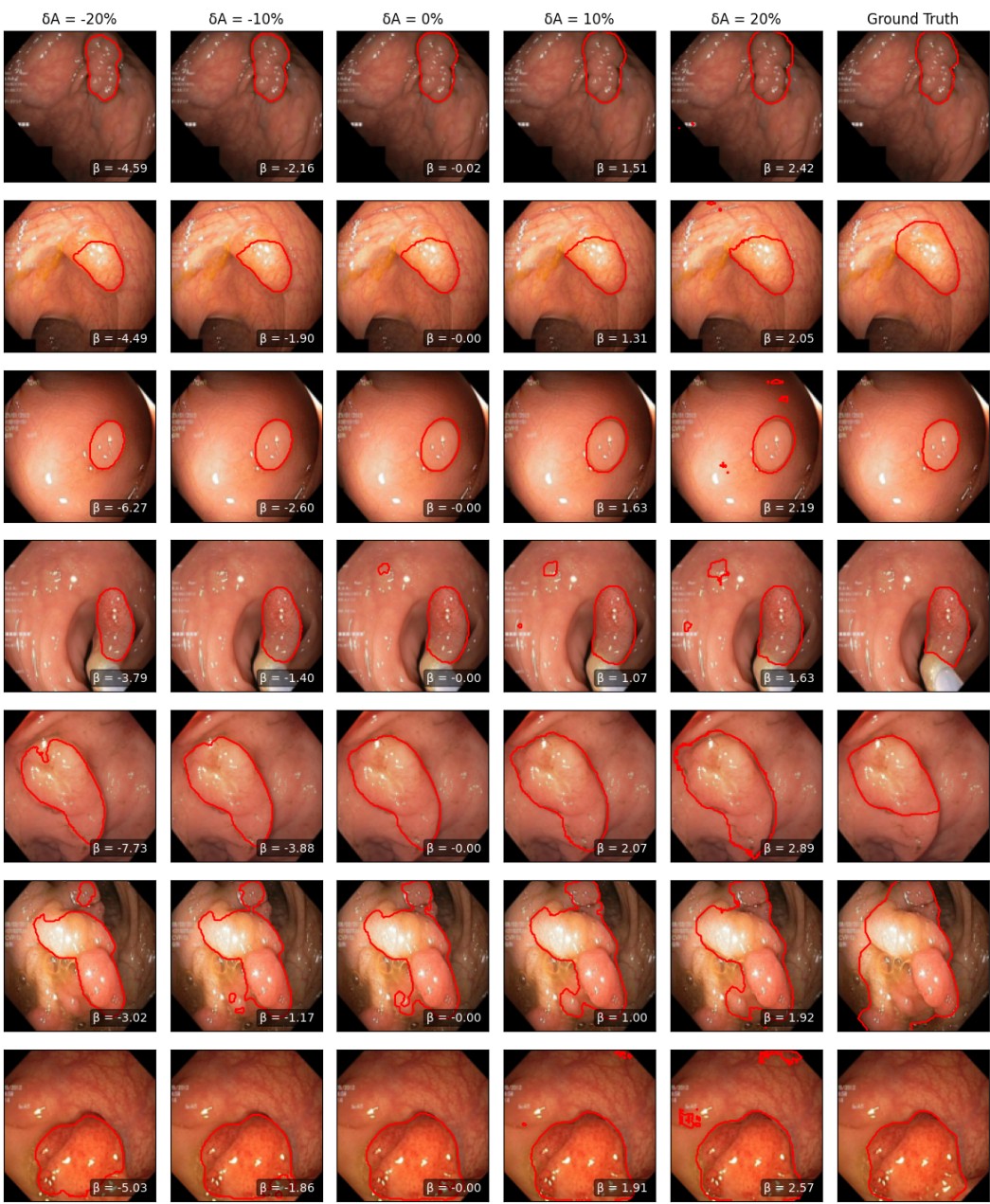

Figure 16: Segmentation area increases with $\beta$ for PolyP. Each row is a different sample. $\delta A = 0\%$ is the original prediction.

### G.2 SEGMENTATION OVERLAYS FOR DIFFERENT PERCENTAGES OF CHANGE IN SEGMENTATION AREA ($\delta A$) FOR COMPASS-L

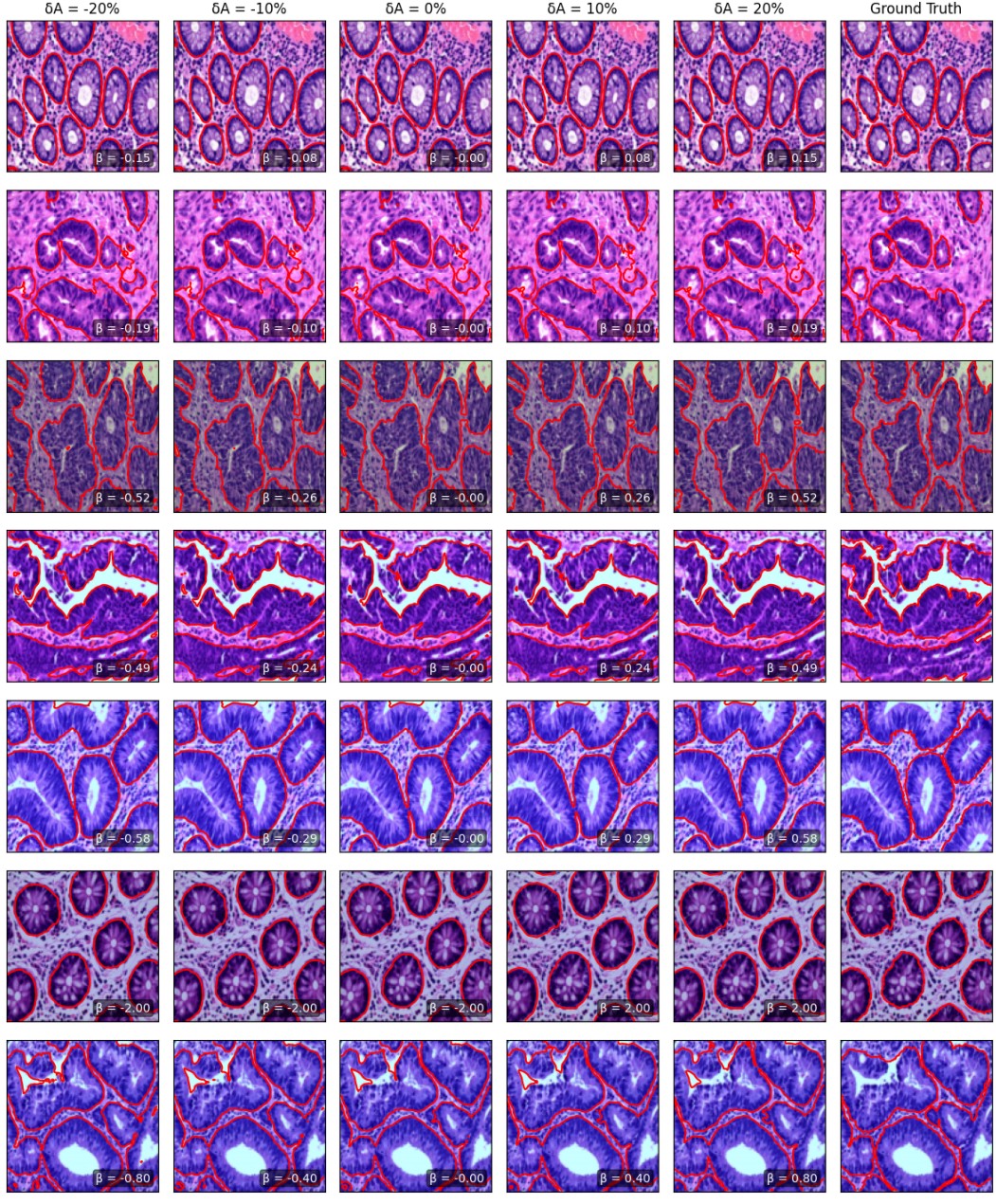

Figure 17: Segmentation area increases with $\beta$ for H&E. Each row is a different sample. $\delta A = 0\%$ is the original prediction.

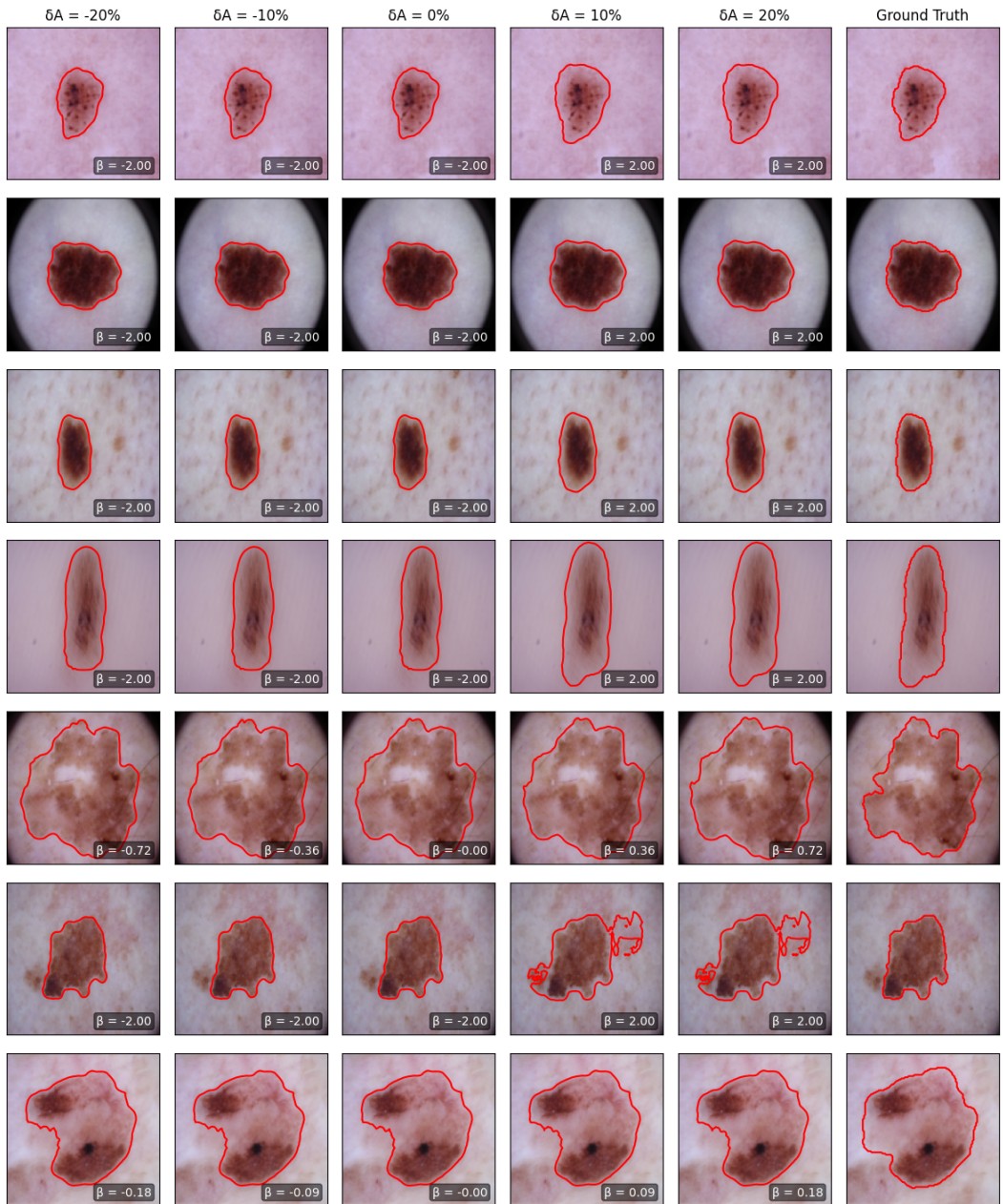

Figure 18: Segmentation area increases with $\beta$ for Skin Lesion. Each row is a different sample. $\delta A = 0\%$ is the original prediction.

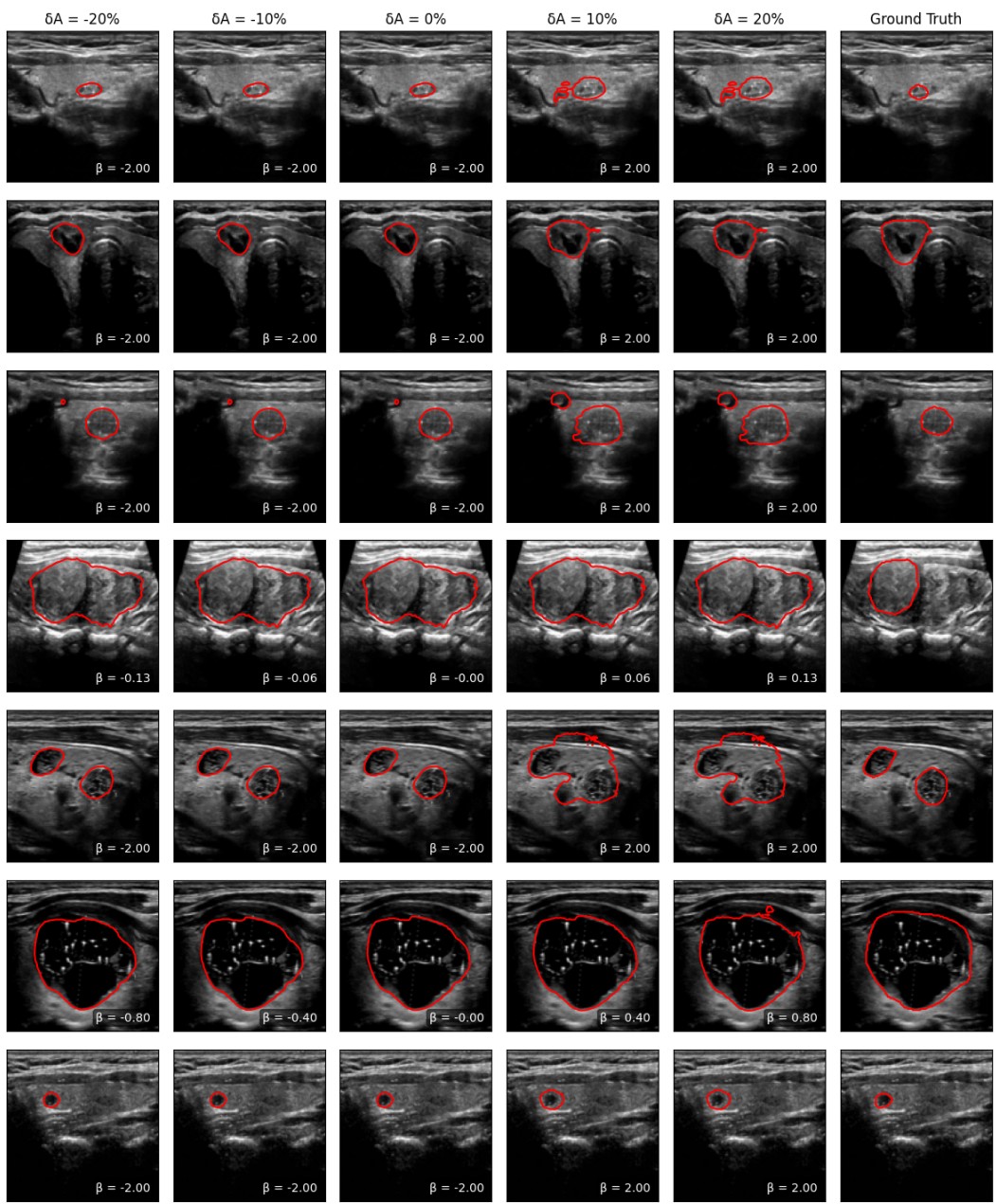

Figure 19: Segmentation area increases with $\beta$ for Thyroid Nodule. Each row is a different sample. $\delta A = 0\%$ is the original prediction.

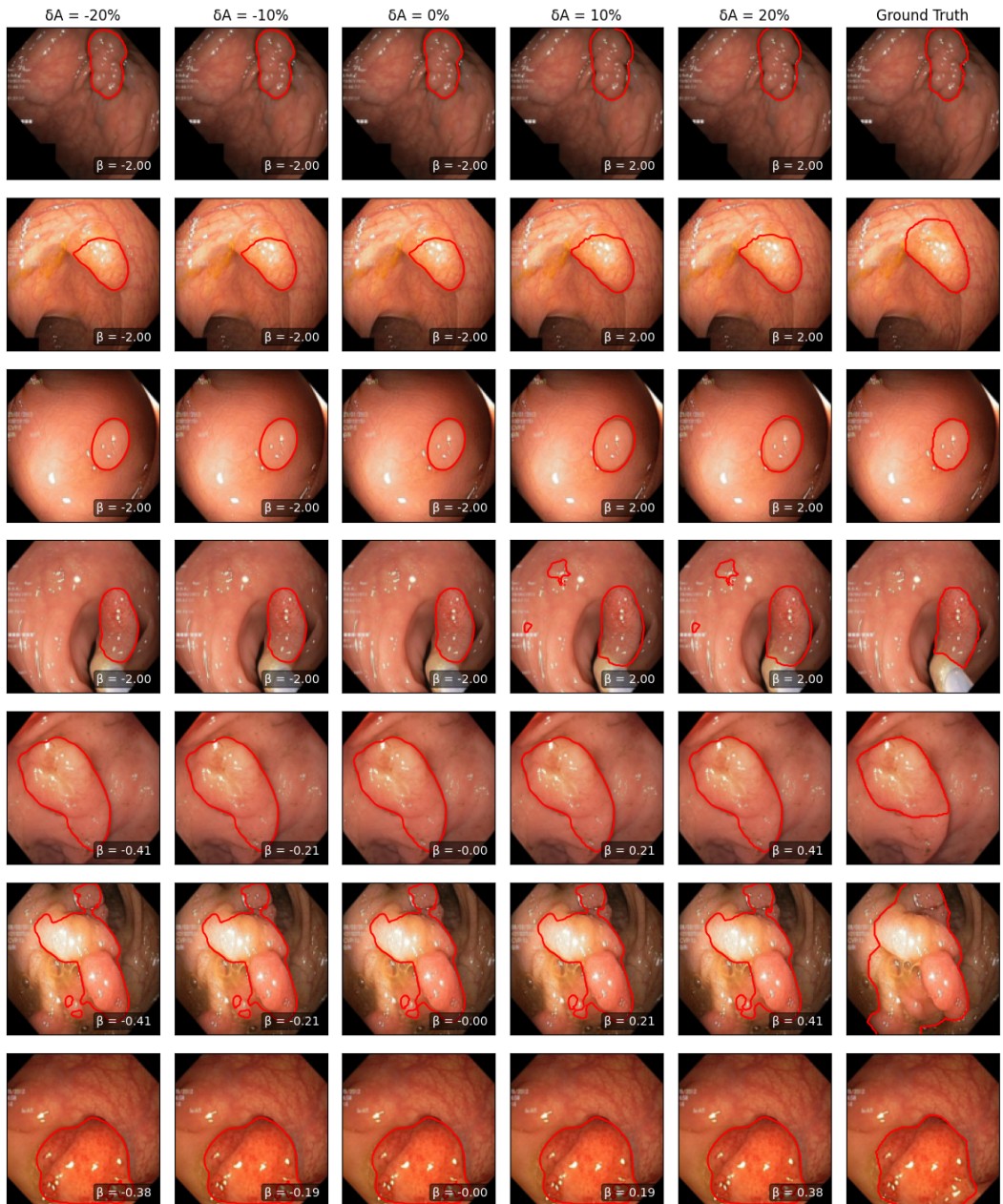

Figure 20: Segmentation area increases with $\beta$ for PolyP. Each row is a different sample. $\delta A = 0\%$ is the original prediction.

