# OpenReview forum: "COMPASS: Robust Feature Conformal Prediction for Medical Segmentation Metrics"
_ICLR.cc/2026/Conference — ICLR 2026 Poster_

### Official Review · Reviewer_ypjS · 2025-10-24

**Soundness:** 3
**Presentation:** 3
**Contribution:** 3
**Rating:** 6
**Confidence:** 3

**Summary:**

The paper targets uncertainty intervals on downstream clinical metrics derived from segmentations, rather than pixel-wise errors. It proposes COMPASS: apply linear perturbations along the most metric-sensitive low-dimensional directions in a network’s intermediate representation, and use the perturbation magnitude as the nonconformity score for split CP to obtain tighter intervals.
Under exchangeability and a nestedness assumption, the method provides marginal coverage guarantees; density-ratio–weighted CP is used to recover target coverage under covariate shift. Empirically, across four medical segmentation tasks, COMPASS (especially the Jacobian variant) achieves shorter intervals while maintaining coverage, and is closer to target coverage under distribution shift.

**Strengths:**

1.	Replaces per-sample adversarial search with a global low-dimensional “sensitivity subspace,” substantially reducing the computational burden of FCP; practically deployable.
2.	Provides standard split-CP coverage under a nestedness condition, and naturally extends WCP with feature/gradient-based weighting to handle coverage under shift.
3.	Demonstrates weighted CP effectiveness under adversarial label/class shift.
4.	Offers empirical evidence for “why it is more efficient”: a power-law relation between score and error with tail compression, plus visualizations showing monotone metric changes under latent-space directional perturbations.

**Weaknesses:**

1. Nestedness assumption is strong and insufficiently vetted. For intervals S_β (x_i )=⋯(Section2.3), monotonicity/nestedness in beta is nontrivial with deep networks and nonlinear metrics. The main text lacks a systematic diagnostic/guarantee mechanism (e.g., monotonicity tests, conservative envelopes, or projection fixes), which is critical for theoretical validity.
2. Feature subspace construction and robustness. Collapsing Jacobians by channel-summing before PCA may sacrifice spatial sensitivity; the choice of layer L, cross-layer comparisons, and PCA stability under small samples/noisy gradients are under-analyzed. A single global subspace may underfit heterogeneous distributions (different anatomies/modalities). Do we need grouped/stratified subspaces or online adaptation?
3. Baselines and scope. The main text omits feasible approximations/variants of FCP; include a practically runnable approximate FCP or feature-perturbation baseline for a fair comparison. Only area (a differentiable metric) is evaluated; adaptation and experiments for nonsmooth/compositional metrics (perimeter, Hausdorff distance, morphology indices, or multi-metric joints) are missing.
4. Coverage guarantee depends on fixed Delta learned only from training data. If Delta adapts using calibration or test distributions, do we need data splitting or multiplicity adjustments to preserve validity?

**Questions:**

1. How do you detect nestedness of S_β (x_i ) in practice? If responses to ±β are asymmetric or non-monotone, what is the remedy—monotone regression, direction re-estimation, or constructing a conservative upper envelope to retain coverage?
2. What is the variance of the PCA subspace across random splits/initializations? Would subspace bagging reduce variance and improve coverage stability?
3. Beyond a LiRPA oracle, can you include approximate adversarial search or low-rank projections + Lp balls as simplified FCP to compare under matched compute budgets?
4. For tiny targets or filamentary anatomy (very small areas), are the intervals still stable? Please report coverage and interval length stratified by target size.
5. If commercial black-box models expose only logits, what are the gains of COMPASS-L versus COMPASS-J? Please specify the minimal API needed (which gradients/intermediate layers) and provide a deployment guide.

**Details Of Ethics Concerns:**

Methodological work on public datasets; while fairness is discussed, no obvious ethics-review triggers.

---

> ### Author Response · Authors · 2025-11-22
> **Author's Response (Part 1)**
>
> Thank you for insightful feedback! We are glad you thought that our paper was sound and has good contributions. While we cannot fully address all of them within the limited rebuttal time window, we have done our best to address the concerns raised and provided multiple additional results to enhance the quality of our submission.
>
> > Nestedness assumption is strong and insufficiently vetted. For intervals S_β(x_i)=⋯(Section2.3), monotonicity/nestedness in beta is nontrivial with deep networks and nonlinear metrics. The main text lacks a systematic diagnostic/guarantee mechanism (e.g., monotonicity tests, conservative envelopes, or projection fixes), which is critical for theoretical validity.
>
> > How do you detect nestedness of S_β (x_i ) in practice? If responses to ±β are asymmetric or non-monotone, what is the remedy—monotone regression, direction re-estimation, or constructing a conservative upper envelope to retain coverage?
>
> **TL;DR:** We agree that there is a lack of theoretical guarantees on the monotonicity of these sets. Theoretically, we adopted your suggestion, and now, nestedness is guaranteed by construction with a conservative envelope. Practically, we provide considerations when using COMPASS (Section 4.1).
>
> These are excellent points. Nestedness is not, and may not hold in practice as constructed in the initial submission. Furthermore, another reviewer also stated that the nestedness assumption in Theorem 1 could be seen as vacuous. We agree with all these points and have now fully addressed this by restructuring our methodology, making our main theoretical guarantee non-vacuous. Please refer to the new sections 2.1 and 2.3. Here is a summary of the key changes:
> 1. **We rigorously redefined the prediction set:** We adopted the suggestion from Reviewer ypjS to use a conservative envelope. Instead of relying on the simple, non-nested endpoint interval $[m_x(-\beta), m_x(+\beta)]$, we have formally redefined the prediction set $S_\beta(x)$ in Theorem 1 as the full range of the metric over the perturbation interval: $$S_\beta(x) := \left[ \min_{b \in [-\beta, +\beta]} m_x(b), \max_{b \in [-\beta, +\beta]} m_x(b) \right]$$
> 2. **We made nestedness a provable property:** With this new min/max definition, nestedness is no longer an assumption but a provable property of our construction. If $\beta_1 \le \beta_2$, the interval $[-\beta_1, +\beta_1]$ is a subset of $[-\beta_2, +\beta_2]$. The min over the larger set must be less than or equal to the min over the smaller set (and the max must be greater than or equal), thus guaranteeing $S_{\beta_1}(x) \subseteq S_{\beta_2}(x)$. This change directly resolves the vacuous nature of the original theorem. The standard conformal proof now holds under the assumption of exchangeability alone.
> 3. **We provided an explicit justification for our practical method:** We bridge the gap between using a conservative envelope and using the endpoints by showing that the two expressions become mathematically equivalent when the metric response function $m_x(b)$ is monotonic. Empirically, our model-metric pairs (UNet and segmentation area) have strongly monotonic metric responses (Figures 2 and 6). This validates our use of the endpoint version of COMPASS as a correct, non-vacuous, and computationally efficient implementation of our framework for this problem class.
> 4. **We updated the proofs to reflect this updated framework:** Specifically, we revised the formal proofs for both the symmetric (Theorem 1) and asymmetric (Theorem 2) cases in the appendix to incorporate the new, more rigorous definition.
>
> We can implement COMPASS in two ways. Again, the two expressions become mathematically equivalent when the metric response function $m_x(b)$ is monotonic. Therefore, our experimental results are justified by the unified Theorem 1:
> 1. **Conservative envelope (non-monotonic):** If the metric response is non-monotonic, computing the extrema requires a full discretized sweep across the perturbation range. A perturbation sweep is required to repeatedly compute the envelope during calibration.
> 2. **Endpoints (monotonic):** When $m_{x_i}(b)$ is *monotonic* in $b$, the conservative envelope collapses to evaluations at the *endpoints*. Thus, the interval can be computed with only two forward passes, giving a practical and efficient implementation of COMPASS.
>
> Practically, we can verify monotonicity by performing a perturbation sweep after the training step (fitting the PCA) before the calibration and testing step. As another reviewer correctly pointed out, this was missing in the algorithms section. We added this check in Algorithm 2.

---

> ### Author Response · Authors · 2025-11-22
> **Author's Response (Part 2)**
>
> > Collapsing Jacobians by channel-summing before PCA may sacrifice spatial sensitivity
>
> **TL;DR:** Collapsing the Jacobians by channel-summing before PCA is by design and produces effective global alignment.
>
> This is an excellent point. In short, we recommend summing the spatial dimensions of the Jacobians before applying dimensionality reduction techniques. There are two parts to this answer:
> 1. **Explained variance is a good indicator of monotonicity:** The choice of the feature layer is critical. We find that the optimal layer can be selected empirically by measuring the *explained variance* of the first principal component ($L=1$) of the Jacobians. High explained variance in the first principal component indicates that the local gradients of the target metric are globally aligned across the samples. Consequently, there is a high chance that a linear traversal along this principal direction results in a consistent, monotonic shift in the output metric for the entire dataset, as the projection scalar maintains a consistent sign. As shown in Figure 5, layers closer to the logits capture the most variance related to object size, validating our use of COMPASS-J.
> 2. **Raw flattened features produce low explained variance:** We observe that the explained variance of the Jacobians is significantly higher when computed on spatially aggregated features compared to raw flattened features (Figure 10). This results from the translation-variant nature of the flattened space, where sensitivity is tied to specific pixel coordinates that vary with object position, leading to orthogonal gradients across samples. In contrast, summing the spatial dimensions isolates the channel-wise semantic sensitivity, which remains consistent across samples.
>
> > the choice of L
>
> The number of eigenvectors $L$ controls the dimensionality of the perturbation subspace. We find that for the segmentation area metric, the first principal component already captures over 90% of the metric's variance (Figure 5). Adding more components does not significantly improve interval tightness.
>
> > cross-layer comparisons
>
> **TL;DR:** We prioritized layers based on the PCA Explained Variance of their metric Jacobians. For our UNet experiments, the penultimate and final layers maximized this metric. We found that a single eigenvector ($L=1$) captures >90% of the variance; adding more components ($L>1$) offers negligible benefit to interval efficiency (Figure 9). Furthermore, we found that the penultimate layer provides us with additional efficiency gains compared to the bottleneck, final layer, and logits (Table 4).
>
> The choice of feature layer is critical. We used the penultimate layer of the UNet. We selected the optimal layer empirically by measuring the explained variance of the first principal component of the metric Jacobians. A high explained variance indicates that the local sensitivity gradients of the metric are globally aligned across the dataset. This implies that a single linear direction exists that can consistently control the metric across different samples. We find that layers closer to the output (specifically the final decoder block, which we used for all experiments) capture the most variance related to object size (Figure 5). Earlier layers showed lower explained variance, indicating a misalignment between those features and the geometric metric. We visualize the effect of these learned directions in Figure 2 and Appendix G.
>
> To justify our choice of penultimate layer, we compare the interval lengths when using the bottleneck layer, penultimate layer (deep) or the final layer (shallow) of the UNet versus the logits (Table 4). We find that while for some datasets the efficiency gains are comparable, deep features generally achieve shorter interval lengths.

---

> ### Author Response · Authors · 2025-11-22
> **Author's Response (Part 3)**
>
> > PCA stability under small samples/noisy gradients are under-analyzed. A single global subspace may underfit heterogeneous distributions (different anatomies/modalities). Do we need grouped/stratified subspaces or online adaptation?
> > What is the variance of the PCA subspace across random splits/initializations? Would subspace bagging reduce variance and improve coverage stability?
>
> **TL;DR:** We found the principal subspace to be highly stable across splits because it captures fundamental geometric properties (scaling) rather than semantic minutiae. While grouped subspaces are a valid extension for multimodal data, our variance analysis suggests bagging is unnecessary for our current setup.
>
> We agree that for highly heterogeneous datasets, a single global PCA subspace could underfit. In such cases, we recommend COMPASS to use conditional subspaces: performing PCA separately for clusters of data (e.g., per-class or per-anatomy). However, for single-task medical segmentation, we found the global subspace effective.
>
> We analyzed the variance of the PCA subspace across random training splits and found that the first principal component ($L=1$) consistently captures almost all of the explained variance (Figure 5) and corresponds to a fundamental geometric transformation (expanding/shrinking the object). Thus, the learned direction is remarkably stable. Given this high stability, subspace bagging such as averaging eigenvectors from bootstrapped splits would increase computational cost during training with negligible gain in coverage stability.
>
> > Only area (a differentiable metric) is evaluated; adaptation and experiments for nonsmooth/compositional metrics (perimeter, Hausdorff distance, morphology indices, or multi-metric joints) are missing.
>
> **TL;DR:** We clarify that area is also natively non-differentiable (discrete pixel count), and we successfully handled it using a differentiable surrogate for direction finding while calibrating on the exact metric. This same protocol applies directly to hausdorff distance (via soft hausdorff) and Perimeter (via boundary loss).
>
> We agree that clinical evaluation often requires diverse metrics. While we are time limited and cannot provide experiments for these, we provide an outline on how to extend COMPASS to these scenarios. Our framework is designed to handle non-smooth and compositional metrics through two specific mechanisms:
> 1. **Differentiable metrics:** Strictly speaking, the clinical metric (pixel count) is discrete and non-differentiable. We bridged this gap by using a differentiable surrogate (Soft Area) only to compute the Jacobian for direction finding. This approach generalizes directly to shape and morphology metrics. For hausdorff distance or perimeter, one would compute $\Delta$ using established differentiable proxies such as soft hausdorff loss or boundary Loss Crucially, the calibration step (Algorithms 3 and 4) requires only function evaluations, not gradients. Therefore, we calibrate on the exact, non-smooth metric (e.g., the true Hausdorff distance). As long as the surrogate-derived direction induces a monotonic change in the true metric (verified by our Algorithm 2), the coverage guarantee holds for the exact clinical metric.
> 2. **Compositional/Joint Metrics:** For multi-metric joints, standard conformal intersection methods apply. We can calibrate $\hat{\beta}_{area}$ and $\hat{\beta}_{shape}$ independently (using a Bonferroni correction) and define the validity set as the intersection of the two intervals. The COMPASS monotonicity property simplifies this, as the intersection of two monotonic interval sets is computationally tractable.
> 3. **Extension to generative models for more complex metrics:**  While standard CNNs have inductive biases for area/object size, COMPASS is well-positioned for emerging architectures designed with semantic alignment in mind. Recent work in diffusion models have constructed “h-spaces” [1,2] that possess strict homogeneity and linearity properties for semantic attributes. These spaces fit perfectly into the COMPASS framework (where $\Delta$ would correspond to a linear traversal in h-space), suggesting that COMPASS will only become more effective as foundational models become more disentangled. We leave this for future work.
>
> [1] Kwon, Mingi, Jaeseok Jeong, and Youngjung Uh. "Diffusion Models Already Have A Semantic Latent Space." The Eleventh International Conference on Learning Representations.
>
> [2] Haas, René, et al. "Discovering interpretable directions in the semantic latent space of diffusion models." 2024 IEEE 18th International Conference on Automatic Face and Gesture Recognition (FG). IEEE, 2024.

---

> ### Author Response · Authors · 2025-11-22
> **Author's Response (Part 4)**
>
> > Coverage guarantee depends on fixed Delta learned only from training data. If Delta adapts using calibration or test distributions, do we need data splitting or multiplicity adjustments to preserve validity?
>
> **TL;DR:** If the subspace $V_L$ adapts using calibration data, data splitting is required to preserve exchangeability. However, we define $\Delta$ as an instance-specific projection onto a fixed training subspace, so no splitting is required, and this works quite well based on our experimental results.
>
> It is crucial to distinguish between the Sensitivity Subspace ($V_L$) and the Instance Direction ($\Delta_i$). In our proposed COMPASS-J, the subspace basis $V_L$ is learned only from $D_{tr}$ and is fixed. For a test point $x_{n+1}$, the direction $\Delta_{n+1} = V_L V_L^T J_{n+1}$ is data-dependent, but the function computing it is fixed. This preserves standard exchangeability, so no additional splitting is needed.
>
> If we were to adapt the basis $V_L$ using the calibration set (e.g., updating PCA components based on $D_{cal}$), the non-conformity scores would become dependent on the entire calibration set, violating exchangeability. In this adaptive case, we would indeed require data splitting: dividing $D_{cal}$ into a some subspace estimation set to update $V_L$ and a score calibration set to compute $R_i$, or utilizing transductive CP, which is computationally more expensive.
>
> > The main text omits feasible approximations/variants of FCP; include a practically runnable approximate FCP or feature-perturbation baseline for a fair comparison.
>
> > Beyond a LiRPA oracle, can you include approximate adversarial search or low-rank projections + $\mathcal{L_p}$ balls as simplified FCP to compare under matched compute budgets?
>
> Thank you for the suggestions. For context, we explored several variants of FCP for segmentation but were unable to obtain a stable or computationally feasible implementation. We first attempted a direct adversarial search (as you propose) over the full feature space (dimensions 64x64x64). This proved computationally intractable: the extremely high dimensionality led to frequent non-convergence of the inner maximization and, as a result, many incorrect nonconformity scores. To partially mitigate this, we collapsed the spatial dimensions to obtain a 64-dimensional feature vector – the same reduced representation used for the reported LiRPA oracle. However, even in this lower-dimensional setting, the adversarial optimization frequently failed to converge, again producing invalid scores and preventing us from achieving the target coverage.
>
> Regarding your suggestion of using low-rank projections together with an $\mathcal{L}_p$​ ball, the approach encounters similar issues. LiRPA cannot be directly applied in this setting because the propagated bounds must correspond to a norm ball that expands uniformly in all directions, whereas a low-rank projection breaks this isotropy. Implementing this variant would therefore require Monte Carlo sampling to approximate the worst-case perturbation, which substantially increases computation while still suffering from non-convergence and instability.
> Ultimately, we opted to report only the oracle variant, in which we directly search over an $\mathcal{L}_p$​ ball for the smallest $\varepsilon$ that expands the predicted segmentation sufficiently to achieve the target coverage. The core advantage of COMPASS is that it uses the Jacobian to explicitly relate the downstream metric to the latent features, drastically reducing the effective search space and avoiding the optimization pathologies encountered in the high-dimensional adversarial and low-rank approaches.
>
> > For tiny targets or filamentary anatomy (very small areas), are the intervals still stable? Please report coverage and interval length stratified by target size.
>
> Thank you for this question. Generally, COMPASS results in stable calibration. To verify this, we we use the results from $\alpha$=0.1, binned the ground truth values into 5 non-overlapping bins, and computed the mean interval length and mean coverage and their 95% confidence intervals in Figure 7. We find that COMPASS methods tend to perform better for shorter interval lengths (except for H&E), achieving approximately the same coverage. The count of the bins (Figure 8) tends to be a better indicator of calibration stability, consistent with other methods.

---

> ### Author Response · Authors · 2025-11-22
> **Author's Response (Part 5)**
>
> > If commercial black-box models expose only logits, what are the gains of COMPASS-L versus COMPASS-J? Please specify the minimal API needed (which gradients/intermediate layers) and provide a deployment guide.
>
> **TL;DR:** For strict black-box models (logits only), COMPASS-L is the only applicable method. It requires zero gradients or internal access. Empirically, COMPASS-L offers slightly lower efficiency gains of COMPASS-J (depending on which dataset). However, COMPASS-L offers significant efficiency gains compared to baselines (Table 1). COMPASS-J requires gray-box access (gradients for training, feature injection for inference) and is best suited for scenarios where interval tightness is paramount :
> 1. **COMPASS-L vs -J:** If a model only exposes logits, COMPASS-L is the designated solution.COMPASS-L (Logit Perturbation): For our setup, COMPASS-L is highly effective despite its simplicity. COMPASS-L significantly outperforms standard baselines and matches COMPASS-J performance on the H&E dataset, albeit slightly less efficient for others. It essentially performs an adaptive level-set expansion based on the model's confidence topography. The gain of using COMPASS-J is strictly statistical efficiency. By exploiting semantic directions in deeper layers, COMPASS-J produced the tightest intervals in 3 of 4 datasets (Skin, Nodule, Polyp), reducing interval width compared to COMPASS-L
> 2. **Deployment:** To deploy these methods, the commercial provider must expose specific endpoints.
>     * Strict Black-Box: The only supported method would be COMPASS-L, requiring the full tensor of logits (pre-softmax). The required input would be the original image only. Based on my understanding, this is supported by almost all standard inference APIs.
>     * Gray-Box / Model Hooks: This would support both COMPASS-L and COMPASS-J. The requirements would be access to gradients to learn the subspace $V_L$. This usually requires white-box offline access once. The API must allow hooks and interrupted executions to extract features and perturb them. This would most likely require custom API endpoints or local model access.
>
> Minor Comments: We fixed a bug in our code and reran our evaluations for Table 1 and 4. The results slightly changed, but the main takeaways remain the same.
>
> Thank you again for all the feedback. We hope these comments and additional results help address your concerns!

---

### Official Review · Reviewer_TJ6a · 2025-10-26

**Soundness:** 3
**Presentation:** 3
**Contribution:** 3
**Rating:** 6
**Confidence:** 4

**Summary:**

The submission focuses on metric-guided conformal prediction for image segmentation models with applications in medical imaging. The submission proposes to construct prediction sets by perturbing the feature representations of pretrained segmentation models along the PCA directions of the gradients of the downstream metric of interest. The method is extended to conformal prediction under distribution shifts.

Experiments compare the proposed method with several existing baselines across a variety of medical imaging datasets.

**Strengths:**

* The paper is well-written.
* Metric-guided conformal prediction is a relevant topic.
* The proposed method is novel, and experimental evaluation is comprehensive.

**Weaknesses:**

* Lack of theoretical guarantees on the monotonicity of the sets makes theorems vacuous for the proposed method in a general sense.
* Method is limited to functions of the segmentation mask only, instead of mask and input.

I have a few clarifying questions and comments and I am looking forward to discussing with the authors!

**Questions:**

**Missing citation to Belhasin et al. (2023)**

The idea of using principal directions to construct prediction sets was introduced by [1] in the context of imaging inverse problems. Although the current submission studies a different problem, it will be important to include this reference.

**Monotonicity of the downstream metric**

The submission acknowledges that nestedness and monotonicity are assumptions of the theoretical results. However, the proposed algorithm does not satisfy these assumptions in practice. Even if this does not seem to affect the experimental results, it does render the theorems vacuous for the proposed method in a general sense. Figure 6 provides evidence that the principal directions do approximate monotonic changes in the downstream metric.

Could the authors expand on technical solutions to guarantee monotonicity? If not, the claim that validity of the algorithm follows by Theorem 1 should be reworded.

**Extending approach to downstream functions of both the image and the mask**

The submission focuses on one downstream task, segmentation area, that only depends on the segmentation mask. Could the authors expand on potential limitations and hurdles to extend the proposed method to:

1. Multi-class segmentation problems.

2. Downstream metrics that also depend on the input image and not the segmentation mask only? This might align better with the motivating example of radiomics in the introduction.

**Dimension of $J_i$**

Could the authors clarify what $C,H$ and $W$ represent? I assume these are not the input image number of channels and spatial dimensions, but the size of the feature embeddings at the chosen layer? Are these vectors flattened in practice to compute the principal directions?

**Experiments**

How many eigenvectors where chosen? Does performance of the proposed method change with $L$?

Which layer was used to compute the directions? Does this choice matter? It could be interesting to include visualizations of some directions.

In Figure 3, why do unweighted CP methods outperform weighted ones?

**Intuition behind Sec. 3.3**

I am not sure I understand the intuitive explanation given in Sec. 3.3. I agree that a skewed distribution has a lower quantile, but how does a lower quantile imply smaller interval lengths here? These quantiles are computed along different directions, which are then pushed forward through another, potentially nonlinear function?

---

**Minor comments**

- It might be worth citing a recent work by Mossina and Friedrich, "Conformal Prediction for Image Segmentation
Using Morphological Prediction Sets" (2025).
- Theorem 1: technically, differentiability does not matter for the theoretical result? If I understand correctly, it is needed to compute the principal directions only.
- Line 175: "optimal" in what sense?

---

**References**

[1] Belhasin et al. "Principal uncertainty quantification with spatial correlation for image restoration problems" (2023).

---

> ### Author Response · Authors · 2025-11-22
> **Author's Response (Part 1)**
>
> Thank you for insightful feedback! We are glad you thought that our paper tackled a relevant topic, proposed method is novel, and the evaluation is comprehensive. We address the concerns raised and provided multiple additional results to enhance the quality of our submission.
>
> > Lack of theoretical guarantees on the monotonicity of the sets makes theorems vacuous for the proposed method in a general sense.
>
> > The submission acknowledges that nestedness and monotonicity are assumptions of the theoretical results. However, the proposed algorithm does not satisfy these assumptions in practice. Even if this does not seem to affect the experimental results, it does render the theorems vacuous for the proposed method in a general sense. Figure 6 provides evidence that the principal directions do approximate monotonic changes in the downstream metric. Could the authors expand on technical solutions to guarantee monotonicity? If not, the claim that validity of the algorithm follows by Theorem 1 should be reworded.
>
> **TL;DR:** We agree that there is lack of theoretical guarantees on the monotonicity of these sets. Theoretically, we adopted a suggestion from Reviewer ypjS and now nestedness is guaranteed by construction. Practically, we provide considerations when using COMPASS (Section 4.1).
>
> These are excellent points. Exchangeability is a standard assumption for CP, but nestedness is not, and may not hold in practice as constructed in the initial submission. Furthermore, another reviewer also stated that the nestedness assumption in Theorem 1 could be seen as vacuous. We agree with all these points and have now fully addressed this by restructuring our methodology, making our main theoretical guarantee non-vacuous. Please refer to the new sections 2.1 and 2.3. Here is a summary of the key changes:
> 1. **We rigorously redefined the prediction set:** We adopted the suggestion from Reviewer ypjS to use a conservative envelope. Instead of relying on the simple, non-nested endpoint interval $[m_x(-\beta), m_x(+\beta)]$, we have formally redefined the prediction set $S_\beta(x)$ in Theorem 1 as the full range of the metric over the perturbation interval: $$S_\beta(x) := \left[ \min_{b \in [-\beta, +\beta]} m_x(b), \max_{b \in [-\beta, +\beta]} m_x(b) \right]$$
> 2. **We made nestedness a provable property:** With this new min/max definition, nestedness is no longer an assumption but a provable property of our construction. If $\beta_1 \le \beta_2$, the interval $[-\beta_1, +\beta_1]$ is a subset of $[-\beta_2, +\beta_2]$. The min over the larger set must be less than or equal to the min over the smaller set (and the max must be greater than or equal), thus guaranteeing $S_{\beta_1}(x) \subseteq S_{\beta_2}(x)$. This change directly resolves the vacuous nature of the original theorem. The standard conformal proof now holds under the assumption of exchangeability alone.
> 3. **We provided an explicit justification for our practical method:** We bridge the gap between using a conservative envelope and using the endpoints by showing that the two expressions become mathematically equivalent when the metric response function $m_x(b)$ is monotonic. Empirically, our model-metric pairs (UNet and segmentation area) have strongly monotonic metric responses (Figures 2 and 6). This validates our use of the endpoint version of COMPASS as a correct, non-vacuous, and computationally efficient implementation of our framework for this problem class.
> 4. **We updated the proofs to reflect this updated framework:** Specifically, we revised the formal proofs for both the symmetric (Theorem 1) and asymmetric (Theorem 2) cases in the appendix to incorporate the new, more rigorous definition.
>
> We can implement COMPASS in two ways. Again, the two expressions become mathematically equivalent when the metric response function $m_x(b)$ is monotonic. Therefore, our experimental results are justified by the unified Theorem 1:
> 1. **Conservative envelope (non-monotonic):** If the metric response is non-monotonic, computing the extrema requires a full discretized sweep across the perturbation range. A perturbation sweep is required to repeatedly compute the envelope during calibration.
> 2. **Endpoints (monotonic):** When $m_{x_i}(b)$ is monotonic in $b$, the conservative envelope collapses to evaluations at the endpoints. Thus, the interval can be computed with only two forward passes, giving a practical and efficient implementation of COMPASS.
>
> Practically, we can verify monotonicity by performing a perturbation sweep after the training step (fitting the PCA) before the calibration and testing step. As you correctly pointed out, this was missing in the algorithms section. We added this check in Algorithm 2.

---

> ### Author Response · Authors · 2025-11-22
> **Author's Response (Part 2)**
>
> > Method is limited to functions of the segmentation mask only, instead of mask and input.
>
> > The submission focuses on one downstream task, segmentation area, that only depends on the segmentation mask. Could the authors expand on potential limitations and hurdles to extend the proposed method to: Multi-class segmentation problems. Downstream metrics that also depend on the input image and not the segmentation mask only? This might align better with the motivating example of radiomics in the introduction.
>
> **TL;DR:** COMPASS is fully compatible with both extensions. Multi-class problems can be handled by calculating a specific $\Delta$ for each class. Input-dependent metrics can be handled naturally because the input image acts as a constant weight map during the Jacobian computation. Since the mask determines which pixels are included, perturbing the latent features could modulate these radiomic features.
>
> We appreciate the suggestion to connect our framework back to the broader radiomics motivation. The mathematical formulation of COMPASS is not limited to geometric properties of the mask.
> 1. **Extension to Input-Dependent Metrics:** COMPASS-J is immediately applicable. The Jacobian calculation requires $\nabla_{\hat{z}} h(g(\hat{z}), x)$ where $x$ is the input image. Since the input image is fixed for a given sample, it functions mathematically as a pixel-wise weight map during the backward pass. For a metric like mean intensity of a region, the function is essentially a weighted average of pixel intensities where the weights are the softmax probabilities. Perturbing $\hat{z}$ along $\Delta$ changes the softmax probabilities (the mask boundaries), which determines which image pixels are included in the average. The gradient signal $\nabla_{\hat{z}} h$ effectively finds the latent direction that expands/shrinks the mask to include/exclude high-intensity regions. Like area, we would need to implement a differentiable version of the radiomic feature to compute the Jacobian for finding $\Delta$, while using the exact metric for the calibration step. A potential hurdle is that since the input image $x$ is fixed, the range of achievable metric values is bounded by the image's intensity distribution. If the ground truth metric lies outside this reachable range (e.g., due to artifacts), the interval effectively saturates. In such cases, a generative approach capable of modulating image content would be required.
> 2. **Extension to Multi-Class Settings:** COMPASS can naturally be applied for multi-class settings by calculating a specific $\Delta$ for each class. Calibration would then produce a vector of perturbation metrics, ensuring marginal coverage for the metric of each class independently. While we do not face significant hurdles as long as the features align with the metrics, a more challenging goal is to achieve simultaneous coverage (i.e., guarantee that 90% of samples have their segmentation metrics simultaneously covered for all classes). Achieving this requires incorporating a multiple testing correction (like Bonferroni) into the calibration step.
> 3. **Other interesting extensions:** Recent work in diffusion models have constructed “h-spaces” [1,2] that possess strict homogeneity and linearity properties for semantic attributes. These spaces fit perfectly into the COMPASS framework (where $\Delta$ would correspond to a linear traversal in h-space), suggesting that COMPASS will only become more effective as foundational models become more disentangled.
>
> [1] Kwon, Mingi, Jaeseok Jeong, and Youngjung Uh. "Diffusion Models Already Have A Semantic Latent Space." The Eleventh International Conference on Learning Representations.
>
> [2] Haas, René, et al. "Discovering interpretable directions in the semantic latent space of diffusion models." 2024 IEEE 18th International Conference on Automatic Face and Gesture Recognition (FG). IEEE, 2024.
>
> > Missing citation to Belhasin et al. (2023). The idea of using principal directions to construct prediction sets was introduced by [1] in the context of imaging inverse problems. Although the current submission studies a different problem, it will be important to include this reference.
>
> Thank you for your suggestion. We have included it in the manuscript. PUQ takes PCA across flattened images stacked together. While PUQ produces tight intervals for pixel-based image reconstruction tasks, its goal is orthogonal to COMPASS, which is to produce intervals for downstream metrics. The use of PCA is in the image space and not the Jacobian of the feature space summed in the spatial dimension (as in COMPASS).

---

> ### Author Response · Authors · 2025-11-22
> **Author's Response (Part 3)**
>
> > Dimension of $J_i$. Could the authors clarify what C, H and W represent? I assume these are not the input image number of channels and spatial dimensions, but the size of the feature embeddings at the chosen layer? Are these vectors flattened in practice to compute the principal directions?
>
> We apologize for any confusion regarding this. Yes, C, H, and W represent the channels, height, and width of the feature embeddings in the chosen layer. The Jacobian tensors are summed in the spatial (H and W) dimensions to get a vector of size C. Then, we proceed to do PCA on the stacked vectors of size [number of data samples, C]. We realized this was not clear in the initial submission. Therefore, we have updated the notation in Section 2.2.
>
> > Which layer was used to compute the directions? Does this choice matter? It could be interesting to include visualizations of some directions.
>
> > How many eigenvectors where chosen? Does performance of the proposed method change with $L$?
>
> **TL;DR:**  We prioritized layers based on the PCA Explained Variance of their metric Jacobians. For our UNet experiments, the penultimate layer maximized this metric. We found that a single eigenvector ($L=1$) captures >90% of the variance; adding more components ($L>1$) offers negligible benefit to interval efficiency (Figure 9).
>
> Thank you for highlighting these important design choices.
> 1. **Layer choice:** The choice of feature layer is critical. We used the penultimate layer of the UNet. We selected the optimal layer empirically by measuring the explained variance of the first principal component of the metric Jacobians. A high explained variance indicates that the local sensitivity gradients of the metric are globally aligned across the dataset. This implies that a single linear direction exists that can consistently control the metric across different samples. We find that layers closer to the output (specifically the final decoder block, which we used for all experiments) capture the most variance related to object size (Figure 5). Earlier layers showed lower explained variance, indicating a misalignment between those features and the geometric metric. We visualize the effect of these learned directions in Figure 2 and Appendix G. We also provide a performance comparison between layers in Table 4.
> 2. **Number of Eigenvectors:** We found that the dimensionality of the sensitivity subspace is very low for the area metric. As shown in Figure 5 the first principal component alone captures over 90% of the variance in the metric Jacobians. Consequently, increasing $L$ beyond 1 does not significantly improve the tightness of the prediction intervals (Figure 9). Therefore, we recommend and use $L=1$ for this task, as it maximizes computational efficiency without sacrificing performance. More complex metrics (e.g., shape or texture), may necessitate $L>1$.
>
> These have been clarified in the updated manuscript.
>
> > In Figure 3, why do unweighted CP methods outperform weighted ones?
>
> For H&E, the unweighted CP methods outperform the weighted ones because they exhibit undercoverage. The weighted CP methods restore this coverage and therefore have larger interval lengths. On the other hand, for Skin Lesion, the unweighted CP methods perform worse than the weighted ones because they exhibit overcoverage. The weighted CP methods restore target coverage and therefore have shorter interval lengths.
>
> > Intuition behind Sec. 3.3. I am not sure I understand the intuitive explanation given in Sec. 3.3. I agree that a skewed distribution has a lower quantile, but how does a lower quantile imply smaller interval lengths here? These quantiles are computed along different directions, which are then pushed forward through another, potentially nonlinear function?
>
> Yes, COMPASS allows the interval to be driven by the local geometry of the test sample (with $\Delta$) and scaled by a $\hat{\beta}$. As shown in Figure 4, the relationship between latent perturbation and output change is sub-linear (concave). To fix a big error in output space, you do not need a proportionally massive perturbation in latent space compared to calibrating in the output space. The model's non-linearity allows small latent shifts to correct large output errors. This mapping effectively squashes the tail end of the distribution. Therefore, the conformalized quantile $\hat{\beta}$ remains moderate. Because $\hat{\beta}$ was determined by a compressed distribution, it is smaller than the equivalent additive expansion factor derived from output errors.
>
> > It might be worth citing a recent work by Mossina and Friedrich, "Conformal Prediction for Image Segmentation Using Morphological Prediction Sets" (2025).
>
> Thank you for this suggestion. We have included this in the introduction.

---

> ### Author Response · Authors · 2025-11-22
> **Author's Response (Part 4)**
>
> > Theorem 1: technically, differentiability does not matter for the theoretical result? If I understand correctly, it is needed to compute the principal directions only.
>
> Thank you for clarifying. Yes, differentiability is not required for the theoretical result, but it is required for COMPASS-J.
>
> > Line 175: "optimal" in what sense?
>
> **TL;DR:** By "optimal," we refer to the adversarial perturbation within a bounded latent neighborhood that induces the maximum possible deviation in the downstream metric. Finding this global maximum is intractable due to the non-convexity of the decoding network.
>
> We define an "optimal" perturbation in the context of the baseline FCP framework as the specific vector within an $\epsilon$-ball in latent space that maximizes the error of the downstream metric. The search for this vector involves a non-convex optimization problem involving the entire decoder network $g$. A naive search (e.g., random sampling or simple gradient ascent like projected gradient descent) is computationally expensive and prone to getting stuck in local optima. If the search yields a sub-optimal $\delta$ (local maximum), the resulting non-conformity score is underestimated, leading to invalid coverage. This intractability is the primary motivation for COMPASS, which replaces the adversarial search with a closed-form projection along semantically dominant directions.
>
> Minor Comments:
> We fixed a bug in our code and reran our evaluations for Table 1 and 4. The results slightly changed, but the main takeaways remain the same.
>
> Thank you again for all the feedback. We hope these comments and additional results help address your concerns!

---

> > ### Comment · Reviewer_TJ6a · 2025-11-26
> > **Thank you for your response!**
> >
> > I sincerely thank the authors for their careful consideration of all reviewers comments.
> >
> > The updated formulation of the prediction sets resolved my theoretical questions, and the updated manuscript clarifies my main questions. I updated my score accordingly.

---

### Official Review · Reviewer_n3H6 · 2025-10-27

**Soundness:** 3
**Presentation:** 3
**Contribution:** 3
**Rating:** 4
**Confidence:** 4

**Summary:**

This paper introduces COMPASS, applying SCP to medical image segmentation tasks, specifically targeting uncertainty quantification for derived metrics like organ size. COMPASS leverages the inductive biases inherent in deep neural networks by performing calibration in the model’s representation space. The method perturbs intermediate features along low-dimensional subspaces that are most sensitive to the target metric. COMPASS yields tighter, more efficient prediction intervals compared to traditional CP methods.

**Strengths:**

1. COMPASS calibrates uncertainty in the model’s intermediate feature space, rather than directly on the output metric. This leverages the inductive biases of deep neural networks, resulting in more efficient and tighter prediction intervals
2. COMPASS incorporates a weighted variant to address distribution shifts.

**Weaknesses:**

1. The method assumes that perturbations along certain directions in the feature space will produce monotonic changes in the metric, this assumption may not hold in all cases, particularly in highly non-linear models, which could lead to suboptimal intervals or coverage guarantees in some real-world scenarios.
2. the weighted COMPASS variant helps correct for moderate distribution shifts, but the validity of the coverage guarantee relies on accurately estimating the density ratio for the calibration and test sets, and errors in this estimation could compromise the method’s effectiveness in practice.
3. COMPASS's performance is heavily reliant on the quality of the pre-trained model’s feature representations. If the model’s features are poorly aligned with the downstream metric, the perturbation directions may become inefficient or non-monotonic, leading to invalid or wide prediction intervals.

**Questions:**

- The paper mentions using PCA to select sensitive perturbation directions, which might be computationally intractable in high-dimensional feature spaces. Please provide further discussion or proof that this method remains efficient in very deep or high-dimensional neural networks
- Theoretically, COMPASS provides valid coverage guarantees under exchangeability and nestedness assumptions. However, in real-world applications, these assumptions might not always hold. how the method ensures valid coverage in scenarios where these assumptions do not fully apply?

---

> ### Author Response · Authors · 2025-11-22
> **Author's Response (Part 1)**
>
> Thank you for insightful feedback! We are glad you thought that our paper was sound and has good contributions. We address the concerns raised and provided multiple additional results to enhance the quality of our submission.
>
> > COMPASS's performance is heavily reliant on the quality of the pre-trained model’s feature representations. If the model’s features are poorly aligned with the downstream metric, the perturbation directions may become inefficient or non-monotonic, leading to invalid or wide prediction intervals.
>
> **TL;DR:** We agree that feature alignment is critical for efficiency, but we clarify that poor alignment does not lead to invalid intervals under our revised framework. We assess alignment using PCA Explained Variance (sensitivity) and Monotonicity Verification (control). High explained variance and monotonicity ensure tight, efficient intervals.
>
> We agree that the efficacy of COMPASS-J depends on the existence of a latent subspace that is sensitive to the target metric. However, we distinguish between the impact on validity versus efficiency:
> 1. **Validity is Guaranteed (and is NOT dependent on features):** Our revised (Theorem 1 guarantees valid marginal coverage regardless of the feature quality or metric non-monotonicity. If features are poorly aligned, the metric response $m_x(b)$ might be flat or erratic. The envelope construction $[\min, \max]$ ensures the set is always nested. As a consequence, the calibration procedure will simply select a very large $\beta$ to satisfy the coverage condition. Thus, poor features lead to wide (inefficient) intervals, but not invalid ones.
> 2. **Efficiency with alignment diagnostics:** To avoid wide intervals, we ensure strong feature alignment using three concrete diagnostics added to our Practical Considerations section:
>     * **Quantifying Alignment (PCA Explained Variance):** The explained variance ratio of the metric Jacobians acts as a proxy for sensitivity. As shown in Figure 5, the first principal component captures almost all of the variance for segmentation area in late layers, indicating strong alignment. Low explained variance would signal a need to switch layers or use COMPASS-L.
>     * **Validating Monotonicity:** We use a perturbation sweep (Algorithm 2) to verify that traversing the principal direction results in a monotonic response. As shown in Figure 2 and 12, our setup exhibits strict monotonicity. This justifies the use of the the endpoint simplification.
> Optimizing layer choice: Alignment is strongest in layers spatially correlated with the output. We recommend selecting the layer that maximizes the PCA explained variance.3.
> Future Architectures: While standard CNNs have inductive biases for area, COMPASS is ideal for emerging architectures with semantic latent spaces. For example, diffusion models with disentangled “h-spaces” [1,2] possess strict homogeneity and linearity, fitting perfectly into our framework.
>
> [1] Kwon, Mingi, Jaeseok Jeong, and Youngjung Uh. "Diffusion Models Already Have A Semantic Latent Space." The Eleventh International Conference on Learning Representations.
>
> [2] Haas, René, et al. "Discovering interpretable directions in the semantic latent space of diffusion models." 2024 IEEE 18th International Conference on Automatic Face and Gesture Recognition (FG). IEEE, 2024.

---

> ### Author Response · Authors · 2025-11-22
> **Author's Response (Part 2)**
>
> > The paper mentions using PCA to select sensitive perturbation directions, which might be computationally intractable in high-dimensional feature spaces. Please provide further discussion or proof that this method remains efficient in very deep or high-dimensional neural networks
>
> **TL;DR:** We mitigate the Jacobian overhead by summing gradients across spatial dimensions before performing PCA, thereby reducing the dimensionality from millions of parameters to a small vector (with a length equal to the number of channels). In our case, this reduces to computing PCA on a [number of data samples, 64] matrix.
>
> We actually find that the PCA of the Jacobians is fast to compute! We achieve this through several techniques and features of COMPASS:
> 1. **Jacobians with respect to metrics:** To clarify, we do not take the Jacobian with respect to the full output space image, which would be extremely expensive. In our experiments with feature spaces of up to 64x64x64, we compute the Jacobians with respect to the metric, which is extremely fast with autograd. This is because the downstream metric is one-dimensional. We present the average computational time to compute the Jacobians for all samples in four datasets below, using a single NVIDIA A100 GPU. We also include this table in the Appendix (Table 8):
> |  | H&E | Skin Lesion | Nodule | PolyP |
> |---|---|---|---|---|
> |Compute time for full dataset (Mean±STD in seconds) | 73.6±0.6 | 305.8±1.1 | 93.3±0.9 | 73.6±0.6 |
> 2. **Summing the spatial dimensions:** We sum the Jacobians in the spatial dimension. This reduction makes the PCA step computationally trivial and independent of the input image resolution, ensuring scalability to 3D volumes or high resolution 2D images. This aggregation aligns with our goal of finding global semantic directions rather than local, pixel-specific adversarial perturbations, allowing us to control the downstream metric monotonically. For the 64x64x64 feature space of the UNet and 2000 samples (for example), we would only need to take the PCA of 2000x64 input, which is a very reasonable size for a classical machine learning problem.
> 3. **Precomputing all the Jacobians:** To expedite the training and calibration step, we precompute and save the Jacobians summed on the spatial dimension before applying PCA.

---

> > ### Author Response · Authors · 2025-11-22
> > **Author's Response (Part 3)**
> >
> > > The method assumes that perturbations along certain directions in the feature space will produce monotonic changes in the metric, this assumption may not hold in all cases, particularly in highly non-linear models, which could lead to suboptimal intervals or coverage guarantees in some real-world scenarios.
> >
> > > Theoretically, COMPASS provides valid coverage guarantees under exchangeability and nestedness assumptions. However, in real-world applications, these assumptions might not always hold. how the method ensures valid coverage in scenarios where these assumptions do not fully apply?
> >
> > > the weighted COMPASS variant helps correct for moderate distribution shifts, but the validity of the coverage guarantee relies on accurately estimating the density ratio for the calibration and test sets, and errors in this estimation could compromise the method’s effectiveness in practice.
> >
> > **TL;DR:** We adopted a suggestion from Reviewer ypjS and now nestedness is guaranteed by construction. Our updated Theorem 1 now only requires exchangeability, which is a standard CP assumption. Our work tackles exchangeability violations by through weighted CP for covariate shift (Section 3.2). Unlike output-space baselines, COMPASS-J grants access to rich, high-dimensional latent representations. We demonstrate that leveraging these discriminative internal features to train the weight estimator (auxiliary classifier) enables highly accurate shift detection, leading to better coverage recovery.
> >
> > These are valid concerns. Exchangeability is a standard assumption for CP, but nestedness is not, and may not hold in practice as constructed in the initial submission. Furthermore, other reviewers have stated that the nestedness assumption in Theorem 1 could be seen as vacuous. We agree with all these points and have now fully addressed this by restructuring our methodology, making our main theoretical guarantee non-vacuous. Please refer to the new sections 2.1 and 2.3. Here is a summary of the key changes:
> > 1. **We rigorously redefined the prediction set:** We adopted the suggestion from Reviewer ypjS to use a conservative envelope. Instead of relying on the simple, non-nested endpoint interval $[m_x(-\beta), m_x(+\beta)]$, we have formally redefined the prediction set $S_\beta(x)$ in Theorem 1 as the full range of the metric over the perturbation interval: $$S_\beta(x) := \left[ \min_{b \in [-\beta, +\beta]} m_x(b), \max_{b \in [-\beta, +\beta]} m_x(b) \right]$$
> > 2. **We made nestedness a provable property:** With this new min/max definition, nestedness is no longer an assumption but a provable property of our construction. If $\beta_1 \le \beta_2$, the interval $[-\beta_1, +\beta_1]$ is a subset of $[-\beta_2, +\beta_2]$. The min over the larger set must be less than or equal to the min over the smaller set (and the max must be greater than or equal), thus guaranteeing $S_{\beta_1}(x) \subseteq S_{\beta_2}(x)$. This change directly resolves the vacuous nature of the original theorem. The standard conformal proof now holds under the assumption of exchangeability alone.
> > 3. **We provided an explicit justification for our practical method:** We bridge the gap between using a conservative envelope and using the endpoints by showing that the two expressions become mathematically equivalent when the metric response function $m_x(b)$ is monotonic. Empirically, our model-metric pairs (UNet and segmentation area) have strongly monotonic metric responses (Figures 2 and 6). This validates our use of the endpoint version of COMPASS as a correct, non-vacuous, and computationally efficient implementation of our framework for this problem class.
> > 4. **We updated the proofs to reflect this updated framework:** Specifically, we revised the formal proofs for both the symmetric (Theorem 1) and asymmetric (Theorem 2) cases in the appendix to incorporate the new, more rigorous definition.
> >
> > **Violations of exchangeability:** We agree that standard CP cannot guarantee coverage when exchangeability is violated. We address this by extending COMPASS to the Weighted CP framework (Section 3.2). In scenarios with covariate shift (e.g., our adversarial hard/easy splits), standard intervals naturally under- or over-cover. We employ weighted CP, which reweights the calibration scores based on the likelihood ratio $w(x) = P_{\text{test}}(x) / P_{\cal}(x)$. We show that COMPASS is fully compatible with this standard remedy. By reweighting the non-conformity scores $R_i$, we restore the exchangeability requirement mathematically.

---

> ### Author Response · Authors · 2025-11-22
> **Author's Response (Part 4)**
>
> **Density ratio estimation accuracy:** We agree that density ratio estimation is a fundamental challenge. However, this is a property of the weighted CP framework itself, not a limitation specific to COMPASS. COMPASS is uniquely positioned to mitigate this risk compared to baselines. Accurate weight estimation depends on the ability to distinguish between calibration and test distributions ($P_{\text{cal}}$ vs. $P_{\text{test}}$). Output-space baselines typically estimate weights using labels etc., which may not be available. These outputs heavily compress information, potentially discarding the very features needed to correct for the shifts. As shown in Figure 3, this allows COMPASS to successfully recover the target coverage even under adversarial shifts, demonstrating predictable and robust behavior. While estimation error is a theoretical risk for any weighted method, COMPASS minimizes this risk practically by providing the weight estimator with the most informative data available (the latent features).
>
> Minor Comments: We fixed a bug in our code and reran our evaluations for Table 1 and 4. The results slightly changed, but the main takeaways remain the same.
>
> Thank you again for all the feedback. We hope these comments and additional results help address your concerns!

---

### Official Review · Reviewer_FBpp · 2025-10-31

**Soundness:** 3
**Presentation:** 3
**Contribution:** 3
**Rating:** 6
**Confidence:** 4

**Summary:**

This paper introduces COMPASS, a novel framework for conformal prediction (CP) tailored to medical image segmentation tasks. Unlike traditional CP methods that operate on scalar outputs or pixel-level segmentation masks, COMPASS generates uncertainty intervals directly for clinically relevant downstream metrics (e.g., organ size) by perturbing intermediate neural network features along directions sensitive to the target metric. The method leverages PCA on Jacobians to identify low-dimensional subspaces that are sensitive to the metrics, enabling efficient and valid interval estimation. COMPASS is evaluated across four medical segmentation datasets and demonstrates superior statistical efficiency and robustness under covariate shift compared to existing CP baselines.

**Strengths:**

1. The paper moves beyond pixel-level CP to metric-level CP, addressing a critical gap in medical image analysis---uncertainty quantification for derived metrics rather than raw segmentation outputs. The method is conceptually good and has the potential to improve trustworthiness in clinical settings.

2. The paper introduces a clever and principled way to perturb latent features along sensitive directions derived from PCA on Jacobians.

3. The authors provide rigorous coverage guarantees under exchangeability and nestedness assumptions.

4. The proposed method produces tighter intervals than existing methods across multiple datasets, maintains valid coverage and adapts well to covariate shifts, while outperforming baselines in both interval efficiency and robustness.

**Weaknesses:**

1. The underlying assumptions---exchangeability and nestedness---may not hold in practice.

2. The method relies heavily on feature quality, i.e., the alignment between extracted features and the evaluation metrics.

3. The experiments focus on segmentation tasks involving area estimation. It remains unclear how well COMPASS generalizes or scales to other metrics (e.g., texture, shape) or modalities (e.g., 3D volumes, multi-class segmentation). In particular, defining a differentiable metric function for such cases may be challenging or even infeasible.

4. The computation of Jacobians can be prohibitively expensive for high-dimensional feature maps. Moreover, the calibration step still requires multiple forward passes, which can be computationally costly for large models or datasets.

**Questions:**

1. How sensitive is COMPASS to violations of the exchangeability and nestedness assumptions? Have you observed practical cases where these assumptions fail, and how does that affect performance or calibration stability?

2. Is there any way to relax or empirically verify these assumptions before applying COMPASS?

3. Since COMPASS relies on the alignment between features and evaluation metrics, how can we assess or quantify this alignment in practice?

4. Would using features from different network layers or models influence the calibration quality of COMPASS? If so, how can one optimally select which layer’s features to use for Jacobian computation?

5. The experiments focus on area-based segmentation metrics—how would COMPASS handle non-differentiable metrics such as texture or shape? Is it feasible to apply COMPASS to 3D volumetric segmentation or multi-class settings, and what challenges would arise?

6. Has any attempt been made to test COMPASS on non-segmentation tasks, such as regression or classification, to evaluate generality?

7. How do you handle the computational overhead of large Jacobians for high-dimensional feature maps?

8. Given that the calibration step requires multiple forward passes, are there strategies to make it more efficient for large-scale datasets or models?

---

> ### Author Response · Authors · 2025-11-22
> **Author's Response (Part 1)**
>
> Thank you for insightful feedback! We are glad you thought that our method was conceptually good, clever, principled, and has the potential to improve trustworthiness in clinical settings. We address the concerns raised and provided multiple additional results to enhance the quality of our submission.
>
> > The underlying assumptions---exchangeability and nestedness---may not hold in practice.
>
> **TL;DR:** Yes, both assumptions were necessary in the original submission, but we adopted a suggestion from Reviewer ypjS and now nestedness is guaranteed by construction.
>
> These are valid concerns. Exchangeability is a standard assumption for CP, but nestedness is not, and may not hold in practice as constructed in the initial submission. Furthermore, other reviewers have stated that the nestedness assumption in Theorem 1 could be seen as vacuous. We agree with all these points and have now fully addressed this by restructuring our methodology, making our main theoretical guarantee non-vacuous. Please refer to the new sections 2.1 and 2.3. Here is a summary of the key changes:
> 1. **We rigorously redefined the prediction set:** We adopted the suggestion from Reviewer ypjS to use a conservative envelope. Instead of relying on the simple, non-nested endpoint interval $[m_x(-\beta), m_x(+\beta)]$, we have formally redefined the prediction set $S_\beta(x)$ in Theorem 1 as the full range of the metric over the perturbation interval: $$S_\beta(x) := \left[ \min_{b \in [-\beta, +\beta]} m_x(b), \max_{b \in [-\beta, +\beta]} m_x(b) \right]$$
> 2. **We made nestedness a provable property:** With this new min/max definition, nestedness is no longer an assumption but a provable property of our construction. If $\beta_1 \le \beta_2$, the interval $[-\beta_1, +\beta_1]$ is a subset of $[-\beta_2, +\beta_2]$. The min over the larger set must be less than or equal to the min over the smaller set (and the max must be greater than or equal), thus guaranteeing $S_{\beta_1}(x) \subseteq S_{\beta_2}(x)$. This change directly resolves the vacuous nature of the original theorem. The standard conformal proof now holds under the assumption of exchangeability alone.
> 3. **We provided an explicit justification for our practical method:** We bridge the gap between using a conservative envelope and using the endpoints by showing that the two expressions become mathematically equivalent when the metric response function $m_x(b)$ is monotonic. Empirically, our model-metric pairs (UNet and segmentation area) have strongly monotonic metric responses (Figures 2 and 6). This validates our use of the endpoint version of COMPASS as a correct, non-vacuous, and computationally efficient implementation of our framework for this problem class.
> 4. **We updated the proofs to reflect this updated framework:** Specifically, we revised the formal proofs for both the symmetric (Theorem 1) and asymmetric (Theorem 2) cases in the appendix to incorporate the new, more rigorous definition.
>
> > Is there any way to relax or empirically verify these assumptions before applying COMPASS?
>
> **TL;DR:** Yes, the new Theorem 1 relaxes this assumption. To test monotonicity of the metric response, we can perform a perturbation sweep before the calibration and testing step.
>
> This is a good question that we had omitted this answer in the initial submission, as pointed out by Reviewer TJ6a as well.
> 1. **Relaxed:** To relax this assumption, we can use the updated Theorem 1, as it guarantees nestedness by definition. If the metric response is non-monotonic, computing the extrema requires a full discretized sweep across the perturbation range. A perturbation sweep is required to repeatedly compute the envelope during calibration.
> 2. **Monotonic:** When $m_{x_i}(b)$ is monotonic in $b$, the conservative envelope collapses to evaluations at the *endpoints*. Thus, the interval can be computed with only two forward passes, giving a practical and efficient implementation of COMPASS.
>
> Practically, we can verify monotonicity by performing a perturbation sweep after the training step (fitting the PCA) before the calibration and testing step. We outline this in Algorithm 2. If the metric response is monotonic, then we can proceed with using the fast endpoint algorithm.

---

> > ### Author Response · Authors · 2025-11-22
> > **Author's Response (Part 2)**
> >
> > > The method relies heavily on feature quality, i.e., the alignment between extracted features and the evaluation metrics.
> > > Since COMPASS relies on the alignment between features and evaluation metrics, how can we assess or quantify this alignment in practice?
> > > Would using features from different network layers or models influence the calibration quality of COMPASS? If so, how can one optimally select which layer’s features to use for Jacobian computation?
> >
> > **TL;DR:** We agree that feature alignment is a prerequisite for effective control. We assess this alignment quantitatively using two specific diagnostics: PCA explained variance to measure sensitivity rank and monotonicity verification to measure predictable control. For our setup, we find that alignment is strongest in layers spatially correlated with the output, and we provide a concrete procedure to select the optimal layer by maximizing the explained variance of the first principal component.
> >
> > We agree that the efficacy of COMPASS-J depends on the existence of a latent subspace that is sensitive to the target metric. We view this as a definable property of the model-metric pair. We provide three concrete ways to assess and optimize this alignment in practice, which we have added to our new Practical Considerations section:
> > 1. **Quantifying alignment with explained variance:** The most direct metric for alignment is the explained variance ratio of the PCA performed on the metric Jacobians. If the first principal component captures (almost) all the variance (Figure 5), there is strong low-rank alignment between the features and the metric. If the explained variance is low or spread across many components, the features are likely not well-aligned with the metric, suggesting a different layer or the logits (COMPASS-L) should be used.
> > 2. **Validating control with perturbation sweep:** While PCA measures sensitivity, it does not guarantee control. We use a perturbation sweep step (Algorithm 2) for the functional assessment of alignment. By performing a perturbation sweep on a small subset of data, users can empirically verify if traversing the principal direction results in a monotonic metric response. As shown in Figure 12 (formerly Figure 6), the UNet-area pair exhibits monotonicity, confirming high-quality alignment.
> > 3. **Optimizing alignment by architecture and layer choice:** Alignment is often strongest in layers spatially correlated with the output. We recommend selecting the layer that maximizes the PCA explained variance (typically layers closer to the output for segmentation tasks). While standard CNNs have inductive biases for area/object size, COMPASS is well-positioned for emerging architectures designed with semantic alignment in mind. For instance, recent works in diffusion models have constructed “h-spaces” [1,2] that possess strict homogeneity and linearity properties for semantic attributes. These spaces fit perfectly into the COMPASS framework (where $\Delta$ would correspond to a linear traversal in h-space), suggesting that COMPASS will only become more effective as foundational models become more disentangled.
> >
> > [1] Kwon, Mingi, Jaeseok Jeong, and Youngjung Uh. "Diffusion Models Already Have A Semantic Latent Space." The Eleventh International Conference on Learning Representations.
> >
> > [2] Haas, René, et al. "Discovering interpretable directions in the semantic latent space of diffusion models." 2024 IEEE 18th International Conference on Automatic Face and Gesture Recognition (FG). IEEE, 2024.

---

> ### Author Response · Authors · 2025-11-22
> **Author's Response (Part 3)**
>
> > The computation of Jacobians can be prohibitively expensive for high-dimensional feature maps. Moreover, the calibration step still requires multiple forward passes, which can be computationally costly for large models or datasets.
> > How do you handle the computational overhead of large Jacobians for high-dimensional feature maps?
> > Given that the calibration step requires multiple forward passes, are there strategies to make it more efficient for large-scale datasets or models?
>
> **TL;DR:** Computing the Jacobians and calibration is actually quite fast (1-5 minutes for whole datasets)! Also, we mitigate the Jacobian overhead by summing gradients across spatial dimensions before performing PCA, thereby reducing the dimensionality from millions of parameters to a small vector (with a length equal to the number of channels). We solve calibration inefficiency by using a binary search, which reduces the process to a logarithmic number of forward passes per sample.
>
> We actually find that the Jacobians and calibrations are quite fast to compute! We achieve this through several techniques and features of COMPASS.
> 1. **Jacobians for metrics:** We do not take the Jacobian with respect to the full output space image, which would be extremely expensive. In our experiments with feature spaces of up to 64x64x64, we compute the Jacobians with respect to the metric, which is extremely fast with autograd. This is because the downstream metric is one-dimensional. We present the average computational time to compute the Jacobians for all samples in four datasets below, using a single NVIDIA A100 GPU. We also include this table in the Appendix (Table 8):
> | | H&E | Skin Lesion | Nodule | PolyP |
> |---|---|---|---|---|
> | Compute time for full dataset (Mean±STD in seconds) | 73.6±0.6 | 305.8±1.1 | 93.3±0.9 | 73.6±0.6 |
> 2. **Summing the spatial dimensions:** We sum the Jacobians in the spatial dimension. This reduction makes the PCA step computationally trivial and independent of the input image resolution, ensuring scalability to 3D volumes or high resolution 2D images. This aggregation aligns with our goal of finding global semantic directions rather than local, pixel-specific adversarial perturbations, allowing us to control the downstream metric monotonically. For the 64x64x64 feature space of the UNet and 2000 samples (for example), we would only need to take the PCA of 2000x64 input, which is a very reasonable size for a classical machine learning problem.
> Precomputing all the Jacobians: To expedite the training and calibration step, we precompute and save the Jacobians summed on the spatial dimension.
> Binary search for $\beta$: We utilize a binary search algorithm rather than a linear scan to compute the non-conformity scores. This reduces the complexity from $\mathcal{O}(N)$ to $\mathcal{O}(\log N)$. For a standard precision, this requires only $\approx$ 10-15 steps per sample, regardless of how large the interval is. We show calibration runtimes in Figure 11.

---

> > ### Author Response · Authors · 2025-11-22
> > **Author's Response (Part 4)**
> >
> > > The experiments focus on segmentation tasks involving area estimation. It remains unclear how well COMPASS generalizes or scales to other metrics (e.g., texture, shape) or modalities (e.g., 3D volumes, multi-class segmentation). In particular, defining a differentiable metric function for such cases may be challenging or even infeasible.
> > > The experiments focus on area-based segmentation metrics—how would COMPASS handle non-differentiable metrics such as texture or shape? Is it feasible to apply COMPASS to 3D volumetric segmentation or multi-class settings, and what challenges would arise?
> >
> > **TL;DR:** COMPASS handles non-differentiable metrics natively without modification. COMPASS-J handles them by using differentiable proxies solely for direction finding. For 3D or multiclass, COMPASS actually scales better than pixel-level baselines because it calibrates a single scalar metric rather than millions of voxels.
> >
> > We appreciate the opportunity to clarify the generalizability of our framework. We focused on segmentation area for our primary clinical demonstration, but the COMPASS framework is adaptable to other metrics and modalities through specific design choices.
> > 1. **Handling non-differentiable Metrics:** Non-differentiable metrics prevent the direct computation of Jacobians for COMPASS-J. For segmentation area, we utilized soft thresholding using the sigmoid function as a differentiable proxy. Furthermore, the calibration and inference algorithms (Algorithms 3 and 4) do not require gradients; they only require evaluating the metric function $h$. Therefore, once the direction $\Delta$ is fixed via the proxy, we perform the calibration search using the exact, non-differentiable metric. The theoretical coverage guarantee (Theorem 1) applies to the metric used during calibration. As long as the surrogate-derived direction $\Delta$ induces a monotonic change in the true metric (a property verified by our Algorithm 2), the framework remains valid and efficient.
> > 2. **Scalability to 3D Volumetric Segmentation:** COMPASS is more scalable to 3D tasks than pixel-level baselines (like SCP) due to being metric-based: The main challenge in 3D is the potential memory cost of computing Jacobians and the increased compute requirement for the forward passes in calibration. However, our Jacobians are computed for the metric, and is computationally cheaper than if the Jacobians for the full image was taken. Furthermore, our method aggregates spatial information before PCA, making the computational cost of COMPASS-J largely independent of the spatial resolution. Once the forward pass is able to be computed, our method uses a binary search instead of linear search to reduce the complexity from $\mathcal{O}(N)$ to $\mathcal{O}(\log N)$. For a standard precision, this requires only $\approx$10-15 steps per sample, regardless of how large the interval is. We show calibration runtimes in Figure 11.
> > 3. **Extension to Multi-Class Settings:** COMPASS can naturally be applied for multi-class settings by calculating a specific $\Delta$ for each class. Calibration would then produce a vector of perturbation metrics, ensuring marginal coverage for the metric of each class independently.

---

> ### Author Response · Authors · 2025-11-22
> **Author's Response (Part 5)**
>
> > How sensitive is COMPASS to violations of the exchangeability and nestedness assumptions? Have you observed practical cases where these assumptions fail, and how does that affect performance or calibration stability?
>
> **TL;DR:** We use weighted CP to tackle exchangeability assumptions and find that COMPASS is able to recover target coverage under covariate shifts (Section 3.2). We guarantee nestedness in our new Theorem 1, so it is no longer a strong assumption.
>
> Exchangeability violations are explored in our work through weighted CP for covariate shift: In standard CP, calibration and testing sets are assumed to be exchangeable. However, when a covariate/distribution shift occurs, these assumptions are violated. This is the premise of Section 3.2, where we show that when covariate shift is present (i.e. increase in hard/easy samples in the test set), the intervals are under-/over-covered. We tackle this exchangeability violation with weighted CP (Section 3.2) by reweighting the non-conformity scores based on the classes, features, or jacobians to restore exchangeability. We show that leveraging learned internal features to estimate importance weights allows COMPASS to recover target coverage under covariate shifts (Figure 3). This result indicates that COMPASS behaves predictably in the presence of exchangeability violations and is compatible with standard remedies like weighted CP.
>
> Nestedness is guaranteed in our updated Theorem 1: For nestedness, as detailed in our updated Theorem 1, we define the prediction set $S_\beta(x)$ as the conservative envelope $[\min m_x(b), \max m_x(b)]$ over the perturbation interval. By definition, expanding the interval $\beta$ cannot shrink the range of the metric; therefore, $S_{\beta_1} \subseteq S_{\beta_2}$ for all $\beta_1 \le \beta_2$. This effectively eliminates sensitivity to nestedness violations. In the hypothetical case where this fails, the remedy is simply to revert to the perturbation sweep to find the extrema in the interval. Practically speaking, if the metric response were non-monotonic, the prediction intervals would need to be inflated.
>
> > Has any attempt been made to test COMPASS on non-segmentation tasks, such as regression or classification, to evaluate generality?
>
> **TL;DR:** We are actively exploring these directions, though they remain outside the scope of the current work.
>
> Thank you for your question. The application of COMPASS to other domains is exciting. While our primary focus was addressing the specific challenges of medical image segmentation (high-dimensional features and geometric metrics), the COMPASS framework is theoretically generalizable to other tasks.
> 1. **Classification:** One could apply COMPASS-J to the penultimate layer features. However, in this setting, the feature space is typically low-dimensional. FCP already works well in this regime because its adversarial search is computationally tractable for low-dimensional vectors. The specific contribution of COMPASS is to solve the tractability gap for high-dimensional spatial feature maps, where standard adversarial search fails. In a classification setting, we expect COMPASS to perform similarly to FCP, whereas in segmentation, COMPASS enables functionality that FCP cannot provide.
> 2. **Image reconstruction:** We see significant potential for COMPASS in high-dimensional regression tasks, such as MRI reconstruction or denoising. The challenge is defining a downstream metric function. For example, if pixel-based intervals are required, then the interval length can serve as the downstream metric. We are actively exploring this research direction for state-of-the-art image reconstruction architectures.
>
> Minor Comments: We fixed a bug in our code and reran our evaluations for Table 1 and 4. The results slightly changed, but the main takeaways remain the same.
>
> Thank you again for all the feedback. We hope these comments and additional results help address your concerns!

---

### Author Response · Authors · 2025-11-30

We sincerely thank the reviewers for their insightful feedback. We would also like to thank the ACs for their tireless efforts during these unprecedented times.

We would like to summarize our progress on rebuttals. In response to the reviewers' comments, we provided a comprehensive point-by-point response to all comments, updated several technical details, and conducted additional experiments to strengthen our work. As a result, 2 out of 4 reviewers have acknowledged this improvement and increased their scores (others were unresponsive):

1. Reviwer FBpp: 6->Unresponsive
2. Reviwer n3H6: 4->6
3. Reviwer TJ6a: 6->8
4. Reviewer ypjS: 6->Unresponsive

We fully understand that the AC has discretion in making decisions, and we provide the above information for reference purposes only. If any clarification is needed, we will be happy to provide it. Thank you!

---

### Meta-Review · Area_Chair_SQJ4 · 2026-01-07

**Summary:**

The submission proposes a conformal prediction framework that works in the latent space of a segmentation neural network.  Experiments demonstrate interval size and coverage indicating that the proposed method can give relatively smaller intervals with similar empirical coverage compared to other conformal prediction variants.

**Reviewer Concerns:**

Concerns included the validity of the assumption of nestedness, or monotonicity of metrics in features.  These are addressed in the rebuttal.  Exchangeability is the remaining assumption.
Other concerns include computational efficiency of the Jacobian (authors responded that this was taken with respect to a 1d metric, so it is efficient), and non-differentiable metrics, which the authors make differentiable with heuristics.
Several reviewers raised concerns about density estimation / validity of coverage.  The authors acknowledged these and indicated it is a general difficulty in conformal prediction and not specific to their method.  Could this explain why target coverage is not reached in a number of reported experiments?

**Reviewer Scores:**

A majority of reviewers initially gave a 6, while one gave a 4.  The technical concerns (nestedness, etc.) seem to largely be addressed, while concerns of empirical coverage remain.

---

### Decision · Program_Chairs · 2026-01-26

Accept (Poster)